# Improving Diffusion Language Model Reasoning through Joint Search in Generation Order and Token Space

## Abstract

The order-agnostic generation of Diffusion Language Models (DLMs) presents a promising alternative to autoregressive models for complex reasoning. We model reasoning as traversals of a problem-specific graph of logical dependencies, and view DLM decoding as sampling trajectories from a joint space over generation orders and token values. We show that standard decoding heuristics such as low-confidence remasking collapse this reasoning space. To address this, we introduce **Order-Token Search**, an algorithm that jointly searches over token content and generation order. Its core is a likelihood estimation function that scores block-level denoising actions, enabling stable path pruning. This allows for efficient exploration of diverse reasoning trajectories. Extensive experiments on mathematical reasoning and planning benchmarks show that our method consistently outperforms baselines, matching or surpassing the gains of fully post-trained d1-LLaDA with *diffu*-GRPO on Countdown, GSM8K, and MATH500 (e.g. achieving a 13.7% absolute gain on Countdown). Our work establishes structured search as a key missing component for advancing reasoning in DLMs.

## 1 Introduction

Recently, Diffusion Language Models (DLMs) have emerged as a powerful alternative to autoregressive (AR) models for sequence generation. A prominent approach, Masked Diffusion Models (MDMs) (Sahoo et al., 2024; Shi et al., 2024), trains on a core objective: learning to reconstruct original text by iteratively denoising sequences where tokens have been randomly masked. At inference, generation begins from a completely masked sequence and proceeds iteratively; the model predicts a full draft, which is then partially **randomly remasked** to form the input for the next denoising step. This training paradigm provides an exploratory objective that fosters order-agnostic generation, contrasting with fixed left-to-right generation, and holds promise for solving complex reasoning tasks that require non-linear thought processes.

The iterative denoising process of MDMs presents a unique opportunity: the choice of which tokens to remask at each step is a free parameter that can be optimized. Rather than relying on **random remasking** used in training, we can guide generation through learned or heuristic remasking strategies. One such strategy, **low-confidence remasking**, leverages the model's uncertainty estimate by locking in high-confidence token predictions as fixed context while remasking low-confidence ones for reconsideration (Nie et al., 2025; Kim et al., 2025a). This prioritizes refinement of uncertain tokens by providing increasingly reliable surrounding context, aiming to improve the model's self-confidence and often leading to higher single-sample performance.

To understand what is gained or lost by fixing a particular remasking strategy, we adopt a task-level view of reasoning. Each problem induces a latent graph of logical dependencies, and any valid solution corresponds to a sequence of intermediate statements that respects this graph. A DLM decoding trace—the sequence of "which position to update" and "which token to place there" across denoising steps—is then one concrete trajectory through this graph. Standard MDM training, however, only directly supervises token predictions under random remasking, leaving the distribution over such trajectories to be determined implicitly by the inference-time remasking rule.

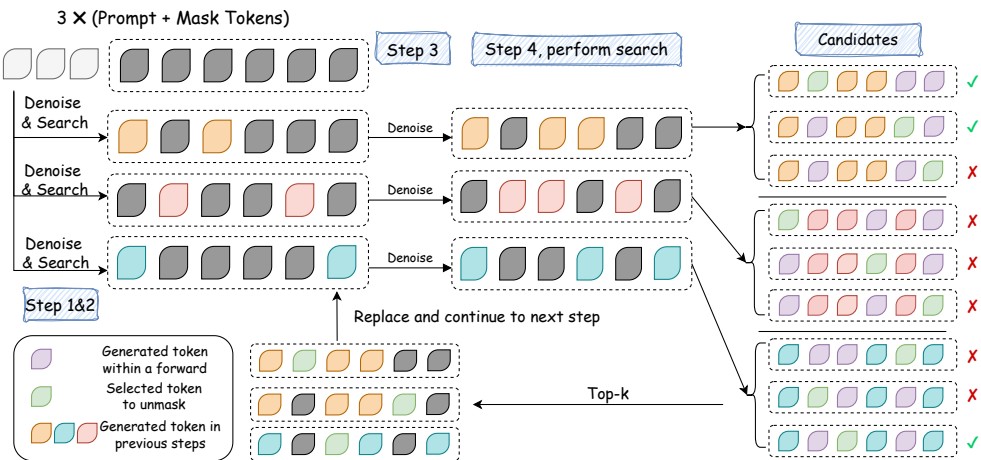

Figure 1: Example of running Order-Token Search algorithm for Diffusion Language Models. Starting from 3 identical fully masked sequences, the reverse diffusion runs for 6 steps to fill in 6 token positions. Every 2 steps (a customizable *search interval*), **the standard denoising is paused, each candidate is expanded into** 3 **candidates, and a sequence-level scoring function is used to prune back to top-**3**.** This process continues to an end where we perform scoring on the top-3 fully denoised sequences to return the optimal one.

While low-confidence remasking improves single-sample accuracy ($pass@1$), we find it inherently limits exploration of potential solutions. We quantify this effect using $pass@k$, the probability that at least one of $k$ samples is correct. Empirically, low-confidence remasking yields superior $pass@1$, but more diverse decoders—such as **random remasking** or a **fixed AR order**—obtain much higher $pass@k$ as $k$ increases by exploring different orders and token choices. This pattern reveals that low-confidence remasking behaves like a greedy search that commits to a narrower set of trajectories in the joint order–token space, whereas diverse strategies expose broader reasoning paths that could reach more correct solutions. Our goal is to search for generation orders that are better aligned with the underlying dependency graph and therefore make the solution logically easy to construct.

We propose Order-Token Search, a new decoding algorithm designed to search in the joint space of generation orders and token choices. Our approach keeps track of multiple candidate sequences (beams) throughout decoding, ultimately returning the one with highest overall generation likelihood. Order-Token Search leverages MDMs' parallel decoding capability—predicting all masked tokens at once. As shown in Figure 1, for each beam, it generates multiple candidate completions for the entire set of remaining masked tokens. These candidates are scored based on sequence likelihood, allowing informed decisions about which paths to pursue.

Through experiments on mathematical reasoning and planning tasks, our method consistently outperforms previous best single-sample decoding (low-confidence remasking). Across Countdown, GSM8K, and MATH500, our test-time search matches or surpasses the gains of fully post-trained d1-LLaDA with *diffu*-GRPO (Zhao et al., 2025); for example, on Countdown it achieves a 13.7% absolute accuracy improvement over the low-confidence baseline. These results demonstrate that explicitly guiding exploration of generation orders and token choices is key to unlocking higher reasoning performance in DLMs.

**Contributions.** We conceptualize reasoning as navigating a graph of logical dependencies, where each problem induces a partial order over intermediate facts and subgoals. This partial order defines a space of valid traversals, while an MDM's denoising trajectory is one particular traversal that may or may not respect these constraints. Low-confidence remasking effectively collapses this space to a single heuristic trajectory, which empirically boosts $pass@1$ but limits $pass@k$ by restricting exploration of alternative orders that can solve more problems (Section 3). In contrast, random remasking explores a much larger portion of the order space, often improving pass@k but at the cost of weaker $pass@1$. To reconcile this trade-off, we introduce Order–Token Search (Section 4), a decoding algorithm that performs structured search over generation orders and token choices,

allowing the model to discover and select trajectories whose generation order better aligns with the underlying logical dependencies. A core technical contribution is a stable likelihood estimation function (Section 4.2) enabling reliable scoring of partial sequences for effective search. Extensive experiments (Section 5) show that this structured exploration yields systematic *pass@1* gains across mathematical and planning benchmarks, matching improvements typically obtained from post-training.

## 2 BACKGROUND

This section establishes the technical foundation for our work. We review the fundamentals of MDMs, formalize key concepts for remasking strategies, and define our evaluation metrics.

### 2.1 DISCRETE DIFFUSION MODELS

Discrete diffusion models adapt the forward diffusion process and the reverse denoising process (Sohl-Dickstein et al., 2015; Ho et al., 2020; Song & Ermon, 2019; Song et al., 2021) to discrete data by establishing the diffusion process over a discrete domain $\mathbf{x} \in \mathcal{X}$, where $\mathbf{x}$ is a one-hot vector denoting tokens from a vocabulary of size $|\mathcal{X}|$ (Austin et al., 2021). Given a prior $\boldsymbol{\pi}$, the forward process $q$ incrementally corrupts the original data $\mathbf{x}_0$ into a target prior distribution $\mathrm{Cat}(\cdot; \boldsymbol{\pi})$. Over continous time $t \in [0, 1]$, it forms a sequence of increasingly noisy latent variables $\mathbf{x}_t$, through the conditional marginal distribution $q(\mathbf{x}_t \mid \mathbf{x}_0) = \mathrm{Cat}(\mathbf{x}_t; \alpha_t \mathbf{x}_0 + (1 - \alpha_t)\boldsymbol{\pi})$. Here, $\alpha_t$ is a monotonically decreasing noise schedule that satisfies boundary conditions $\alpha_0 = 1$ and $\alpha_1 = 0$. Furthermore, we can achieve the transition probability between any two intermediate time points $0 < s < t < 1$ through $q(\mathbf{x}_t \mid \mathbf{x}_s, \mathbf{x}_0) = \mathrm{Cat}(\mathbf{x}_t; \alpha_t/\alpha_s \mathbf{x}_s + (1 - \alpha_t/\alpha_s)\boldsymbol{\pi})$.

MDM, a specific instance of this framework, utilizes the prior $\boldsymbol{\pi} = \boldsymbol{m}$ to achieve absorbing-state diffusion, a particularly suitable setting for language modeling (Sahoo et al., 2024; Shi et al., 2024; Lou et al., 2024). Here, $\boldsymbol{m}$ is a one-hot vector corresponding to a special MASK token. Defining $s$ as the time step immediately preceding $t$, the posterior distribution simplifies to:

$$q(\mathbf{x}_s \mid \mathbf{x}_t, \mathbf{x}_0) = \begin{cases} \mathrm{Cat}(\mathbf{x}_s; \mathbf{x}_t), & \mathbf{x}_t \neq \boldsymbol{m} \\ \mathrm{Cat}\left(\mathbf{x}_s; \frac{\alpha_s - \alpha_t}{1 - \alpha_t}\mathbf{x}_0 + \frac{1 - \alpha_s}{1 - \alpha_t}\boldsymbol{m}\right), & \mathbf{x}_t \neq \boldsymbol{m} \end{cases} \tag{1}$$

The reverse (denoising) process is modeled by $p_\theta(\mathbf{x}_s \mid \mathbf{x}_t) = q(\mathbf{x}_s \mid \mathbf{x}_t, \mathbf{x}_\theta(\mathbf{x}_t))$, where $p_\theta$ is a parameterized distribution that reverses $q$, and $\mathbf{x}_\theta(\mathbf{x}_t)$ denotes a neural network trained to predict the original clean data $\mathbf{x}_0$ from its noisy version $\mathbf{x}_t$. This network is optimized by minimizing the negative evidence lower bound, thereby learning to approximate the true posterior distribution.

### 2.2 REMASKING STRATEGIES IN MDM SAMPLING

In masked generative models, sampling starts from a fully masked sequence, $\mathbf{x}_1 = (\mathrm{MASK}, \ldots, \mathrm{MASK})$. The model then iteratively refines this sequence over a series of steps. At each step, the model predicts logits for all currently masked tokens. The critical action in this reverse process is the transfer of a prediction—that is, the act of replacing a selected MASK token with its predicted value, thereby committing to that prediction for subsequent steps. The rule that determines which masked token to transfer next is known as the **remasking strategy**, and it defines the decoding order. We focus on three primary strategies:

**Random Remasking.** The strategy used during training. The next token to unmask is chosen uniformly at random from the set of all remaining masked tokens. This is a baseline that ensures unbiased, order-agnostic generation. **Autoregressive (AR).** We force the DLM to keep the leftmost predicted token and remask all following tokens. This baseline decouples the effect of generation order and solely examines the effect of diverse token selection. **Low-Confidence Remasking.** A common inference-time strategy. The token with the highest predicted probability is unmasked next; the tokens with lower probability are remasked. Formally, at each step, the model computes a confidence score for each masked token $i$ as its maximum logit, $s_i = \max(p_\theta(\cdot \mid \mathbf{x}_t)_i)$. The token with the *maximum* score $s_i$ is transferred. The intuition is to resolve the token position where the model has the greatest certainty first, potentially mitigating error propagation (Nie et al., 2025; Kim et al., 2025a).

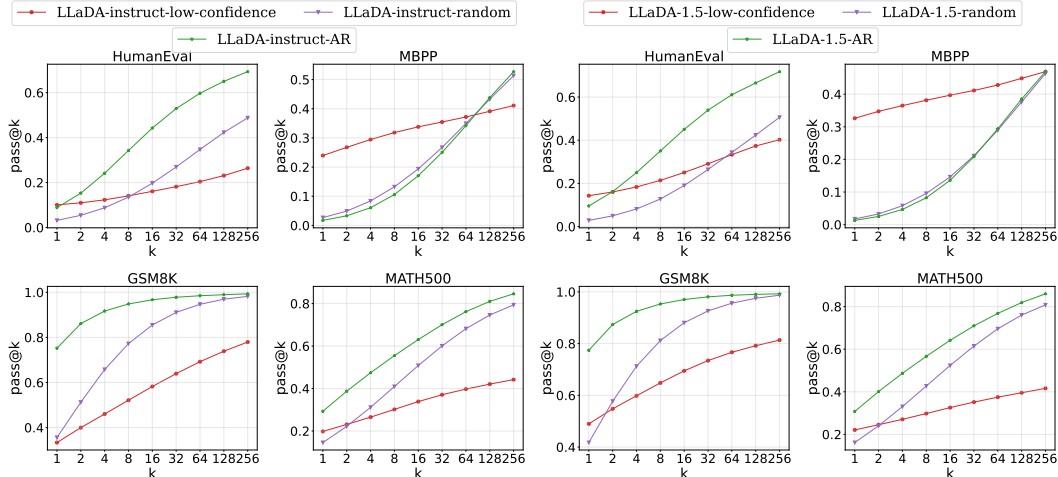

Figure 2: Empirical *pass@k* curves for LLaDA-8B-Instruct and LLaDA-1.5 on reasoning and coding benchmarks. **While low-confidence remasking often achieves higher accuracy (*pass@1*), both random remasking and autoregressive (AR) decoding yield superior *pass@k* for large $k$ (≈256)**, revealing a higher performance potential. This indicates that strategies exploring more diverse paths solve more unique problems overall.

### 2.3 EVALUATION METRICS FOR REASONING PERFORMANCE

Evaluating generative models on reasoning tasks requires metrics that capture both deterministic performance and the model's inherent capability. We use the following standard metrics established in prior work (Yue et al., 2025): **Accuracy (*pass@1*).** The probability that a single generated sample is correct. This is the primary metric for evaluating one-trial performance and represents the expected accuracy when using the model in a deterministic setting. *pass@k.* The probability that at least one sample out of $k$ independent generations is correct. This metric estimates the model's inherent ability to solve a problem given sufficient sampling. For a problem with $n$ generated samples of which $c$ are correct, it is estimated as: $pass@k \approx 1 - \binom{n-c}{k} / \binom{n}{k}$.

The relationship between these metrics reveals important characteristics of a decoding strategy. A strategy with high accuracy but low *pass@k* for large $k$ indicates that while effective for single samples, it under-utilizes the model's true potential by failing to explore diverse solution paths. A core goal of our work is to develop a decoding algorithm that achieves higher accuracy by better exploring the joint space of generation orders and token selections.

## 3 REASONING PERFORMANCE OF STANDARD DECODING STRATEGIES

**Decoding Strategy Trade-offs.** To understand the fundamental trade-offs in DLM decoding strategies, we systematically investigate how different approaches affect reasoning performance by addressing a critical question: *How does the diversity of generation paths explored by a decoding strategy relate to its ability to solve complex problems?* We analyze three core strategies representing distinct exploration-exploitation trade-offs: random remasking (maximizing generation order diversity), low-confidence remasking (greedily exploiting local confidence), and fixed autoregressive order (enforcing generation order while enabling diverse token selection). This comparison is particularly valuable because DLMs offer unique flexibility in generation order compared to autoregressive models, yet optimal strategies for leveraging this flexibility in reasoning tasks remain unclear.

**Low-confidence remasking: high single-sample accuracy but rapid performance plateau.** Figure 2 presents the key findings from our analysis. Using LLaDA models (Nie et al., 2025; Zhu et al., 2025) trained for flexible-order generation, we evaluate each strategy's single-sample accuracy (*pass@1*) and multi-sample coverage (*pass@k*) across mathematical reasoning (GSM8K, MATH500) and coding (HumanEval, MBPP) benchmarks to study the effects of generation order

and token selection diversity on reasoning performance. As expected, the low-confidence remasking strategy achieves the highest $pass@1$ in most cases, with advantages of 0.2-0.3 on MBPP. This aligns with the intuition that resolving the most certain tokens first helps guide the generation and reduces error propagation in a single sample. However, as we increase the sample budget $k$, the performance of low-confidence remasking plateaus relatively quickly.

**Diverse strategies: lower initial accuracy but superior multi-sample coverage.** In contrast, both random remasking and fixed AR order strategies, while starting at lower $pass@1$, continue to improve with more samples, eventually achieving significantly higher $pass@k$ (e.g., random/AR strategies reach ∼0.8 while low-confidence plateaus at ∼0.4 at $pass@256$ on MATH500). This demonstrates that diverse generation orders (from random remasking) and token selections (from AR decoding with temperature) can enable the model to solve more unique problems. We also conducted an experiment (Appendix A.2) showing that AR is not fundamentally superior than other generation orders. This implies that exploration in the generation order and token space, rather than the specific left-to-right order, is what improves the model's reasoning performance. The sets of solutions generated through these diverse paths have greater coverage of the solution space, even though each individual sample is less reliable.

This consistent phenomenon reveals a fundamental limitation of greedy decoding strategies: by committing early to specific token choices based on local confidence, low-confidence remasking restricts exploration of the joint space of generation orders and token selections. While effective for optimizing a single sample, this approach fails to utilize the model's full reasoning potential. The model's inherent capability, as measured by $pass@k$ with large $k$, is better realized by strategies that introduce more stochasticity.

Our analysis reveals the core opportunity in DLM decoding: achieving higher accuracy than low-confidence remasking requires strategic exploration of both generation order and token space. To this end, we introduce Order-Token Search, an algorithm that actively searches this joint space within a single decoding process to locate correct answers that greedy strategies miss.

## 4 METHOD: ORDER-TOKEN SEARCH FOR DIFFUSION LANGUAGE MODELS

The empirical results in Section 3 establish the need for a decoding algorithm that explores multiple potential generation paths. Standard one-trial sampling is insufficient as it commits to a single greedy sequence. To address this, we develop Order-Token Search, a novel algorithm inspired by beam search but tailored for the parallel, iterative nature of DLMs. The key innovation is a joint search over both the **token** selections and the **order** in which they are generated, guided by the model's own likelihood predictions to score and prune candidate paths. This section details its two key components—search and prune. The complete algorithm is provided in Appendix A.1.

### 4.1 SEARCH PROCESS

We begin with $K$ identical copies (beams) of the initial sequence $\mathbf{x}_1 = [\mathbf{c}; \text{MASK}^L]$, where $\mathbf{c}$ is the prompt and $\text{MASK}^L$ denotes $L$ mask tokens. Over continuous denoising time $t \in [0, 1]$, we independently apply the MDM to each candidate, generating new hypotheses that represent different choices in the joint space of tokens and generation orders. Between any two user-specified time $(s, t)$, our algorithm can perform search and expand each candidate to become multiple candidates independently with additional denoise steps. This expansion explores the joint space of tokens and generation orders, creating a diverse set of candidate sequences which are then evaluated and pruned to retain the top-$K$ paths with the highest model likelihood (see Section 4.2).

To manage computational complexity, we structure the search using **block diffusion** (Arriola et al., 2025; Nie et al., 2025). Instead of searching at every denoising step—which would incur an $O(K \cdot |t|)$ overhead for $|t|$ steps—we perform the search expansion only at the boundaries between contiguous blocks of tokens. This reduces the overhead to $O(K \cdot |b|)$, where $|b|$ is the total number of blocks, making the search tractable. After processing all blocks, the single best sequence is selected from the final $K$ candidates based on the highest likelihood. Details on computational complexity are provided in Appendix A.5.

Figure 3: Illustration of a pruning stage in Order-Token Search for DLMs. At a search step, we have 2 fully denoised sequences (on the leftmost), with yellow tokens unmasked in previous steps. **We then mask the current block (the middle 3 tokens) and measure its likelihood through feeding each masked candidate into the DLM to obtain each token's probability.** The score function computes the chain-rule product of token probabilities and prunes the lower-likelihood candidate.

## 4.2 PRUNING PROCESS

The effectiveness of Order-Token Search hinges on a pruning criterion that can accurately score candidate sequences with diverse tokens and generation orders. Our key insight is that the standard MDM objective can be an unreliable scorer, as the model is trained on—and often fails at—extremely difficult infilling tasks where a large number of tokens must be predicted simultaneously (Kim et al., 2025a). To obtain a more stable and accurate likelihood estimate, we instead score a candidate based on the incremental denoising actions that created it.

We propose a scoring function $s$ that evaluates the model's confidence for each discrete denoising step. For a step from a more corrupted state $\mathbf{x}_t$ to a less corrupted state $\mathbf{x}_s$ (where $0 \leq s < t \leq 1$), the score is calculated as:

$$s(\mathbf{x}_t; \mathbf{x}_s) = \mathbb{E}_{\mathbf{x}_0 \sim p_\theta(\mathbf{x}_0|\mathbf{x}_t)} \log p(\mathbf{x}_0 | b(\mathbf{x}_s, \mathbf{x}_t, \mathbf{x}_0)), \tag{2}$$

where $p(\cdot|\cdot)$ is the parametrized posterior from Section 2.1. The function $b$ identifies the specific blocks of token positions $\{i \mid \mathbf{x}_{t,i} = \text{MASK} \cap \mathbf{x}_{s,i} \neq \text{MASK}\}$ that were denoised between time $t$ and $s$, masks these blocks in $\mathbf{x}_0$, and returns the masked sequence. The score $s(\mathbf{x}_t; \mathbf{x}_s)$ is the log-likelihood of *only these newly-revealed blocks*, conditioned on the surrounding context provided by the model's full-sequence prediction $\mathbf{x}_0$.

This approach provides a better likelihood estimation with lower variance. By focusing on smaller, incremental predictions, we assess the model on tasks similar to its well-learned training distribution, where it denoises a limited number of masks at a time. The total score for a candidate sequence is the sum of scores over all its search-guided denoising steps: $\sum_{(s,t) \in \mathcal{I}} s(\mathbf{x}_t; \mathbf{x}_s)$, where $\mathcal{I}$ is the set of intervals where a search was performed. This sum captures the entire history of the candidate's generation path. Figure 3 illustrates this process for a single step.

In summary, Order-Token Search performs a joint search over tokens and generation orders, guided by the MDM's own likelihood. By allowing candidates to explore denoising paths independently, Order-Token Search achieves diverse and effective exploration of the joint output space. The algorithm evaluates this exploration by scoring a candidate's entire generation history through incremental block-level likelihoods, providing a comprehensive measure of global coherence. This approach efficiently leverages the iterative, parallel nature of MDMs: block diffusion minimizes computational overhead, while the full-sequence prediction $\mathbf{x}_0$ supplies rich context for stable likelihood estimation. Consequently, Order-Token Search enables effective pruning of low-likelihood paths, steering the search toward high-quality, coherent outputs.

## 5 EXPERIMENTS

We conduct a series of experiments to evaluate the effectiveness of Order-Token Search in improving the reasoning performance of MDMs. Our investigation centers on the following research questions: (1) Does Order-Token Search yield consistent improvements in reasoning accuracy over competitive

Table 1: **Model performance on Mathematics and Planning Benchmarks.** We report accuracy across four benchmarks and multiple generation lengths for two base models (LLaDA and LLaDA-1.5). Bolded values indicate the best performance. With the stronger AR- and majority-voting-based baselines, no single method dominates every setting; however, Order-Token Search (Order-Token Search) achieves the highest overall accuracy (*All*) for both base models and attains the best dataset-level averages on MATH500 and Countdown, while remaining competitive with the strongest baselines on GSM8K and Sudoku.

| | | All | GSM8K | | | | | MATH500 | | | | | Countdown | | | | | Sudoku | | | | |
|---|---|---|---|---|---|---|---|---|---|---|---|---|---|---|---|---|---|---|---|---|---|---|
| | Method / Seq Len | All | 64 | 128 | 256 | 512 | Avg | 64 | 128 | 256 | 512 | Avg | 64 | 128 | 256 | 512 | Avg | 64 | 128 | 256 | 512 | Avg |
| LLaDA | Low-confidence | 31.1 | 44.3 | 68.7 | 76.7 | 78.2 | 67.0 | 21.2 | 26.0 | 32.4 | 36.2 | 29.0 | 25.8 | 20.7 | 19.5 | 16.0 | 20.5 | 8.5 | 11.7 | 6.5 | 5.5 | 8.1 |
| | Low-conf + MV | 32.5 | 46.4 | 72.5 | 80.9 | 83.1 | **70.7** | 20.2 | 27.4 | 35.0 | 36.2 | 29.7 | 22.7 | 23.8 | 18.4 | 18.0 | 20.7 | 8.9 | 10.3 | 8.5 | 7.2 | 8.7 |
| | Random + MV | 28.8 | 43.1 | 70.7 | 80.2 | 80.3 | 68.6 | 17.2 | 26.2 | 31.8 | 31.8 | 26.8 | 6.3 | 15.2 | 14.1 | 15.2 | 12.7 | 6.4 | 6.5 | 8.4 | 6.9 | 7.1 |
| | Order-Token Search | **35.2** | 45.6 | 71.6 | 79.8 | 83.3 | 70.1 | 22.4 | 30.4 | 36.0 | 42.4 | **32.8** | 27.7 | 34.4 | 26.2 | 25.4 | **28.4** | 10.1 | 11.7 | 8.5 | 7.4 | 9.4 |
| | AR | 28.8 | 34.0 | 62.7 | 75.7 | 76.9 | 62.3 | 18.8 | 23.4 | 27.4 | 34.4 | 26.0 | 10.6 | 12.9 | 13.3 | 14.1 | 12.7 | 14.7 | 15.9 | 13.6 | 12.0 | 14.1 |
| | AR + MV | 31.0 | 41.0 | 69.1 | 81.7 | 86.4 | 69.6 | 17.4 | 23.0 | 32.2 | 39.9 | 28.1 | 10.2 | 13.3 | 11.3 | 13.7 | 12.1 | 14.4 | 15.9 | 12.1 | 14.3 | **14.2** |
| | AR + beam-search | 33.3 | 40.3 | 70.4 | 81.1 | 82.9 | 68.7 | 22.2 | 26.6 | 35.4 | 39.8 | 31.0 | 18.4 | 23.1 | 21.5 | 21.9 | 21.2 | 13.2 | 17.7 | 10.6 | 8.2 | 12.4 |
| LLaDA-1.5 | Low-confidence | 32.3 | 44.2 | 69.6 | 77.5 | 79.4 | 67.7 | 20.2 | 26.4 | 32.5 | 36.2 | 28.8 | 19.8 | 19.7 | 17.9 | 21.8 | 19.8 | 13.5 | 15.6 | 11.4 | 10.3 | 12.7 |
| | Low-conf + MV | 35.0 | 49.1 | 74.4 | 84.1 | 84.2 | **73.0** | 21.2 | 30.0 | 34.8 | 39.3 | 31.3 | 20.7 | 23.8 | 20.3 | 25.4 | 22.5 | 13.7 | 14.9 | 10.8 | 12.6 | 13.0 |
| | Random + MV | 30.2 | 47.7 | 74.8 | 81.2 | 82.8 | 71.6 | 22.4 | 26.2 | 31.0 | 30.4 | 27.5 | 5.5 | 16.4 | 9.0 | 14.5 | 11.4 | 11.1 | 11.3 | 9.3 | 9.2 | 10.2 |
| | Order-Token Search | **36.7** | 48.4 | 74.5 | 81.7 | 84.0 | 72.2 | 24.4 | 30.8 | 37.4 | 42.4 | **33.8** | 27.7 | 31.3 | 23.8 | 29.3 | **28.0** | 13.8 | 16.1 | 11.2 | 9.9 | 12.8 |
| | AR | 30.3 | 37.5 | 67.7 | 77.3 | 79.1 | 65.4 | 17.2 | 23.5 | 31.2 | 35.0 | 26.7 | 12.3 | 15.2 | 14.5 | 15.7 | 14.4 | 14.5 | 15.8 | 14.5 | 13.6 | 14.6 |
| | AR + MV | 32.2 | 40.7 | 72.9 | 83.8 | 84.2 | 70.4 | 18.4 | 26.4 | 33.6 | 37.9 | 29.1 | 12.5 | 17.2 | 14.8 | 16.0 | 15.1 | 14.6 | 16.6 | 13.2 | 13.1 | 14.4 |
| | AR + beam-search | 33.5 | 45.1 | 73.2 | 82.0 | 84.9 | 71.3 | 19.0 | 26.4 | 35.2 | 38.8 | 29.9 | 14.8 | 21.1 | 16.0 | 20.3 | 18.1 | 17.5 | 16.8 | 13.5 | 11.0 | **14.7** |

baselines—such as low-confidence remasking and majority-voting—across a variety of tasks? (2) How does the likelihood estimation of Order-Token Search compare to a naive autoregressive-like approximation? (3) How does performance scale with beam size $K$, and at what point do we observe diminishing returns? Finally, we provide a case study where Order-Token Search successfully solves a problem that other baseline methods fail to resolve.

## 5.1 EXPERIMENTAL SETUP

We compare Order-Token Search against several strong baselines: **Low-confidence remasking**, a greedy decoding method adopted as an optimal base model configuration in Zhao et al. (2025). **Random remasking with majority voting**, which generates a compute-equivalent set of diverse samples via random remasking and selects answers using a consistency heuristic (Wang et al., 2022). **Low-confidence with majority voting**, which combines the greedy decoding with the consistency heuristic mentioned above. **AR**, which follows the left-to-righ autoregressive order in generation. **AR with majority voting** and **AR with beam search**, which strengthen the AR baseline with, respectively, a consistency heuristic and a likelihood-based search. **Order Search**, a computationally expensive algorithm that uses AR-like likelihood to search for the optimal generation order. **Token Search**, an equally expensive algorithm that uses AR-like likelihood to search through the top-$K$ likely tokens for each position.

**Model and Tasks.** Our primary testbed is **LLaDA-8B-Instruct** (Nie et al., 2025), a state-of-the-art open-source diffusion language model. Since it has not undergone post-training with methods like *diffu*-GRPO (Zhao et al., 2025), it offers a clean baseline for isolating the performance improvements attributable to our inference-time algorithm. We additionally evaluate **LLaDA-1.5** (Zhu et al., 2025), an RL post-trained variant of LLaDA, to verify that our conclusions hold even after reinforcement-learning-based post-training. For tasks, we evaluate on two mathematical reasoning and two planning benchmarks. **GSM8K** (Cobbe et al., 2021) contains ∼1.32k grade school math problems requiring multi-step reasoning. **MATH500** (Lightman et al., 2023) is a challenging subset of 500 high-school competition-level problems from the MATH (Hendrycks et al., 2021) dataset. **Countdown** (Pan et al., 2025) is a combinatorial arithmetic game where the goal is to reach a target number using basic operations on a given set. **Sudoku** requires logical reasoning and constraint satisfaction to solve a puzzle grid (Zhao et al., 2025).

## 5.2 ORDER-TOKEN SEARCH IMPROVES REASONING ACCURACY

**Overall performance: Order-Token Search is the strongest decoding across benchmarks.** As shown in Table 1, after adding majority-voting and autoregressive baselines, no single decoding method dominates every dataset or sequence length, but Order-Token Search remains the strongest overall. For both LLaDA and LLaDA-1.5, it attains the highest *All* accuracy (35.2% vs. 33.3% for the best baseline on LLaDA, and 36.7% vs. 35.0% on LLaDA-1.5) and the best dataset-level averages

Table 2: **Accuracy of search algorithms with different likelihood estimate.** Bolded values indicate best performance. Order-Token Search consistently outperforms both Order Search and Token Search that adopt an AR-like likelihood estimate.

| Decoding Method (Compute) | GSM8K | MATH500 | Countdown | Sudoku |
|---|---|---|---|---|
| Token Search (3x) | 8.5 | 3.8 | 0.0 | 6.1 |
| Order Search (3x) | 79.2 | 35.8 | 15.2 | 5.9 |
| Order-Token Search (1x) | **79.8** | **36.0** | **26.2** | **8.5** |

Table 3: On the Countdown task, **Larger beam size** results in **higher accuracy**.

| Beam Size($K$) | Accuracy (%) |
|---|---|
| K=1 | 16.0 |
| K=3 | 19.1 |
| K=5 | 20.3 |
| K=8 | 21.1 |

on MATH500 and Countdown. On GSM8K, diffusion-style decoding (Low-confidence, Low-conf + MV, Random + MV) and autoregressive decoding (AR, AR + MV, AR + beam-search) trade wins with Order-Token Search at different sequence lengths while Order-Token Search remains close in performance, whereas on Sudoku the AR variants are typically strongest, as discussed below.

**Diffusion baselines: Order-Token Search improves over remasking and voting on hard reasoning tasks.** Within diffusion-style decoding, the greedy Low-confidence baseline already provides solid performance, and adding majority voting reliably improves GSM8K averages for both models. Random + MV, which replaces confidence-based remasking with random remasking, can be competitive on GSM8K but substantially degrades the tasks that requires more structured reasoning: its Countdown averages fall to 12.7% and 11.4% for LLaDA and LLaDA-1.5, compared with 20.7% and 22.5% for Low-conf + MV. In contrast, Order-Token Search consistently improves over these diffusion baselines on MATH500 and Countdown: its MATH500 averages reach 32.8% and 33.8% (vs. 29.7% and 31.3% for Low-conf + MV), and its Countdown averages increase to 28.4% and 28.0%. For example, on Countdown with LLaDA at length 128, Order-Token Search attains 34.4% accuracy, versus 23.8% for Low-conf + MV and 15.2% for Random + MV.

**Autoregressive baselines: Order-Token Search outperforms AR on reasoning, while Sudoku is a backbone failure case.** Comparing to autoregressive decoders, AR, AR + MV, and AR + beam-search are strong baselines on GSM8K and Sudoku, with AR + MV and AR + beam-search often achieving the best GSM8K scores at longer sequence lengths and the best Sudoku performance for both backbones. Nevertheless, Order-Token Search still provides clear advantages on the harder reasoning tasks: averaging the dataset-level averages over both LLaDA variants, its GSM8K/MATH500/Countdown scores are 71.2%/33.3%/28.2%, compared to 70.0%/28.6%/13.6% for AR + MV and 70.0%/30.5%/19.7% for AR + beam-search (a `block_size=1` special case of our search). Finally, all Sudoku performance remain far below the 25% accuracy of uniform guessing on a $4 \times 4$ grid (e.g., Order-Token Search at 9.4%/12.8% and AR + beam-search at 12.4%/14.7%), so we view Sudoku as a failure case of this backbone rather than evidence against the value of searching jointly over orders and tokens.

## 5.3 THE NECESSITY OF DEDICATED LIKELIHOOD ESTIMATION FOR MDMS

The efficacy of any search-based decoding algorithm is contingent upon its capacity to accurately estimate the likelihood of candidate sequences for effective pruning. As established in Section 4.2, employing a naive or autoregressive (AR)-style likelihood estimation is suboptimal for MDMs, which inherently model tokens at multiple positions in parallel. Our baseline search algorithms, Order Search and Token Search, utilize an AR-style likelihood estimation by computing the logits of revealed tokens in a forward pass. The inferior performance of these baselines in Table 2, compared to Order-Token Search, provides initial evidence that this scoring method is misaligned with the MDM paradigm.

On the Countdown task, Order-Token Search surpasses Order Search by 11%, while Token Search degrades the base model's performance to 0%. This result substantiates that performing a naive beam search over token values, guided by a sequence of greedily-decided positions (selected via low-confidence remasking), is ineffective for MDMs. Furthermore, searching the generation orders in isolation is insufficient, as Order Search requires triple the computational cost to approach the performance of Order-Token Search. The superior and efficient performance of Order-Token Search is directly attributable to its dedicated likelihood estimation and its joint exploration of the generation order and token space.

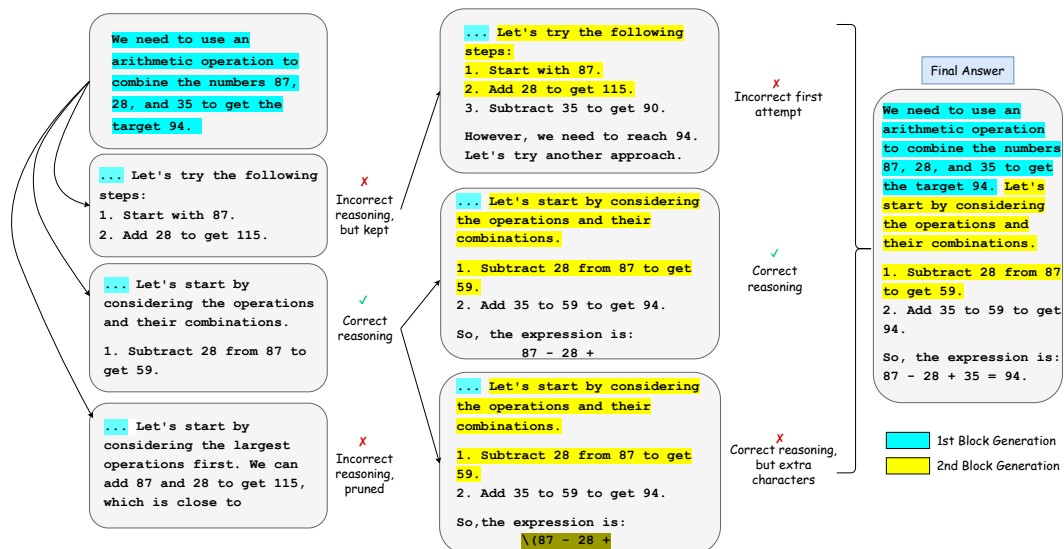

Figure 4: Case study of search trajectories for a sampled Countdown problem. Each box depicts a independently generated candidate sequence with arrows denoting the parent-child relationship in block diffusion. Order-Token Search evaluates each candidate and decides whether to move forward with its prefix sequence. Its likelihood criterion successfully pruned out inferior candidates that contains incorrect reasoning or syntactical errors, ultimately retaining only high-scoring candidates that lead to the correct solution.

## 5.4 SCALING WITH BEAM SIZE

We evaluate the scalability of Order-Token Search by analyzing the trade-off between reasoning accuracy and computational cost as the beam size $K$ increases. Results on the Countdown task (Table 3) show that performance improves consistently with beam size, rising from **16.0%** at $K = 1$ (equivalent to greedy decoding) to **21.1%** at $K = 8$. The marked gain from $K = 1$ to $K = 3$ (+3.1%) confirms that even modest beam expansion helps escape suboptimal greedy paths.

Beyond $K = 3$, however, we observe diminishing returns: accuracy increases by only 1.2% from $K = 3$ to $K = 5$, and 0.8% from $K = 5$ to $K = 8$. These results indicate that while larger beams enable more thorough exploration and higher accuracy, the marginal improvement decreases as $K$ grows. This establishes a practical guideline for setting $K$ to maximize accuracy within a given computational budget.

## 5.5 CASE STUDY: A SEARCH INSTANCE

Figure 4 provides a qualitative analysis of Order-Token Search's search trajectory on a Countdown task requiring the combination of numbers $(87, 28, 35)$ to reach a target value of $94$. This particular problem exemplifies a case where the low-confidence remasking baseline fails, as it greedily commits to the locally plausible but ultimately incorrect path beginning with $87 + 28 = 115$. However, this path cannot yield the target $94$ using the remaining number $35$, since $115 \pm 35$ results in values ($80$ or $150$) distant from the solution.

Order-Token Search overcomes this limitation by maintaining multiple candidate paths simultaneously. While the addition-based path is explored, the algorithm also evaluates the alternative $87 - 28 = 59$ trajectory. The dedicated likelihood estimation correctly identifies the subtraction path as superior when $59 + 35$ precisely yields the target $94$. This case demonstrates how Order-Token Search's joint exploration of generation orders and token space enables escape from local optima that trap greedy methods, systematically identifying globally correct solutions through parallel hypothesis testing.

## 6 RELATED WORK

**Diffusion Language Models.** Initial developments in discrete diffusion models were established by D3PM (Austin et al., 2021) and further progressed using masked token approaches (Sahoo et al., 2024; Nie et al., 2024). Efficient versions such as Plaid (Gulrajani & Hashimoto, 2023) and SEDD (Lou et al., 2024) achieve performance comparable to GPT-2 (Radford et al., 2019), but their scalability still falls short of autoregressive models. The most recent scaling efforts include Dream (Ye et al., 2025), which adapts pre-trained autoregressive models into diffusion models, and LLaDA (Nie et al., 2025), which trains powerful diffusion language models from scratch.

**Test-Time Strategies.** A primary method to enhance diffusion models is to increase test-time compute, often by using more denoising steps. Recent work has shown that expanding the inference-time sample space can guide generation toward high-reward outputs (Singhal et al., 2025; Kim et al., 2025b), with techniques like re-masking being introduced to scale the denoising process for masked diffusion models specifically (Wang et al., 2025). In the broader context of language models, search algorithms like beam search, speculative decoding (Leviathan et al., 2023; Xia et al., 2023), and contrastive decoding (Li et al., 2022) have been developed to improve decoding beyond greedy selection. However, these algorithms are tailored for the autoregressive paradigm, where the search space is confined to token values given a fixed generation order.

Our work addresses this limitation. The iterative denoising of MDMs creates a joint search space over both token values and their generation order, which is inaccessible to autoregressive methods. Our algorithm, Order-Token Search, is designed for this new paradigm, leveraging parallel decoding to explore multiple generation paths and select outputs based on overall likelihood.

## 7 CONCLUSION

In this work, we revisited decoding for Diffusion Language Models through the lens of reasoning. We modeled each problem as inducing a graph of logical dependencies, with any DLM decoding trace corresponding to a particular trajectory through this graph. Our analysis of $pass@k$ revealed that standard low-confidence remasking effectively collapses the rich space of possible trajectories to a narrow set of greedy paths: it improves single-sample accuracy but restricts exploration of alternative generation orders that would solve more problems, whereas more diverse remasking strategies broaden this space at the expense of $pass@1$.

To reconcile this trade-off, we introduced **Order-Token Search**, a decoding algorithm that performs structured search in the joint space of generation orders and token values while leveraging the parallel denoising nature of MDMs. By maintaining multiple candidate hypotheses and guiding the search with a novel, block-based likelihood estimation, Order-Token Search discovers trajectories whose generation orders better align with the underlying dependency graph. Experiments on mathematical reasoning and planning benchmarks show that this inference-time search yields systematic gains in accuracy ($pass@1$), matching or surpassing the improvements of expensive post-training methods such as $diffu$-GRPO. These findings highlight that decoding-time search over orders is not merely a heuristic refinement, but a central ingredient for unlocking the reasoning capabilities latent in DLMs.

## ETHICS STATEMENT

This work adheres to the ICLR Code of Ethics. All experiments are conducted using publicly available benchmark datasets (GSM8K, MATH500, Countdown, and Sudoku) and pretrained models (LLaDA-8B-Instruct, LLaDA-1.5), with our contributions centered on methodological and algorithmic advancements, in particular the development of the **Order-Token Search** method which explores the joint space of generation order and token selection. The study does not involve human subjects, personal data, or other sensitive information, and no applications with a high likelihood of causing harm are considered. We have carefully examined potential risks and broader societal impacts of this research and did not identify significant ethical concerns.

## REPRODUCIBILITY

We have taken deliberate steps to ensure that our work can be reliably reproduced. Detailed descriptions of the datasets and model employed are provided in Section 5.1, while Appendix A.4 outlines the inference setups and hyperparameter configurations. The details of **Order-Token Search** are documented in Appendix A.1. Upon acceptance, we will make our algorithm implementation publicly available to support reproducibility and enable subsequent research.

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

## USE OF LLMs

Large language models (LLMs) were used solely to assist with grammar refinement and writing clarity during the manuscript preparation stage. All technical ideas, experimental designs, model implementations, and analyses were conceived and executed by the authors without reliance on LLMs. The use of LLMs did not influence research outcomes, data interpretation, or reported results. We carefully reviewed and edited all text to ensure accuracy, originality, and compliance with ethical and academic standards.

## A APPENDIX

### A.1 ORDER-TOKEN SEARCH ALGORITHM

To make our proposed decoding strategy more concrete, we present the pseudocode of our Order-Token Search, which illustrates how partially masked sequences are expanded, scored and pruned, exploring both the token space and order space.

---

**Algorithm 1** Order-Token Search for Diffusion Language Models

---

1: **Input**: Prompt $\mathbf{p}$, model $p_\theta$, beam size $K$, generation length $L$, total steps $S$, search interval $N$, temperature $\tau$, number of blocks $b$.
2: Initialize beam set $\mathcal{B} \leftarrow \{(\mathbf{x}_i, \mathbf{s}[b], \text{score})\}$, where $\mathbf{x}_i = [\mathbf{p}; \text{MASK}^L]$, $\mathbf{s}[0 : b - 1] = 0$, score $= 0$             $\triangleright K$ identical beams
3: **for** step $s \leftarrow 1$ to $S$ **do**
4:      $\mathbf{l} \leftarrow p_\theta(\mathcal{B}.\mathbf{x})$           $\triangleright$ Get logits for all beams, shape: $(K, L, V)$
5:      **if** $s \mod N == 0$ **then**             $\triangleright$ Search step
6:          $\mathcal{B}_{\text{candidates}} \leftarrow \emptyset$
7:          **for** $(\mathbf{x}, \mathbf{s}, \text{score}) \in \mathcal{B}$ **do**
8:             block_idx $\leftarrow$ get_current_block_index($\mathbf{x}$)     $\triangleright$ Compatible with semi-AR generation
9:             **for** $i \leftarrow 1$ to $K$ **do**        $\triangleright$ Expand each beam into $K$ candidates
10:                $\tilde{\mathbf{l}} \leftarrow$ add_gumbel_noise($\mathbf{l_x}, \tau$)        $\triangleright$ Perturb logits for exploration
11:                $\mathbf{x}_0 \leftarrow \text{argmax}(\tilde{\mathbf{l}}, \dim = -1)$        $\triangleright$ Sample a candidate completion
12:                $\mathbf{x}_{\text{candidate}} \leftarrow$ transfer_tokens($\mathbf{x}, \mathbf{x}_0, \mathbf{l_x}$)     $\triangleright$ Only apply $\frac{L}{S}$ predicted tokens
13:                $\mathbf{x}_{\text{full\_seq}} \leftarrow$ transfer_all_tokens($\mathbf{x}, \mathbf{x}_0$)     $\triangleright$ Apply all predicted tokens
14:                $\mathbf{x}_{\text{masked}} \leftarrow$ mask_tokens($\mathbf{x}_{\text{full\_seq}}$, block_idx)     $\triangleright$ Mask the current block
15:                block_score $\leftarrow$ score_block($\mathbf{x}_{\text{masked}}$, block_idx)     $\triangleright$ Score the sequence
16:                $\mathbf{s}[\text{block\_idx}] = $ block_score
17:                score $= \text{sum}(\mathbf{s}[0 : \text{block\_idx}])$
18:                $\mathcal{B}_{\text{candidates}} \leftarrow \mathcal{B}_{\text{candidates}} \cup \{(\mathbf{x}_{\text{candidate}}, \mathbf{s}, \text{score})\}$
19:             **end for**
20:          **end for**
21:          $\mathcal{B} \leftarrow \text{top}_K(\mathcal{B}_{\text{candidates}})$        $\triangleright$ Prune to the $K$ best candidates
22:      **else**        $\triangleright$ Standard sampling step
23:          **for** $(\mathbf{x}, \_\_, \_\_) \in \mathcal{B}$ **do**        $\triangleright$ Update each beam independently
24:             $\tilde{\mathbf{l}} \leftarrow$ add_gumbel_noise($\mathbf{l_x}, \tau$)
25:             $\mathbf{x}_0 \leftarrow \text{argmax}(\tilde{\mathbf{l}}, \dim = -1)$
26:             $\mathbf{x} \leftarrow$ transfer_tokens($\mathbf{x}, \mathbf{x}_0, \mathbf{l_x}$)        $\triangleright$ No scoring/pruning
27:          **end for**
28:      **end if**
29: **end for**
30: **Return**: The sequence from $\mathcal{B}$ with the highest final score.

---

### A.2 IS THE AUTOREGRESSIVE ORDER FUNDAMENTALLY SUPERIOR?

The parallel between random remasking and a fixed AR order in Figure 2 raises a natural hypothesis: perhaps the samples that lead to the high *pass@256* for random remasking are those that, by

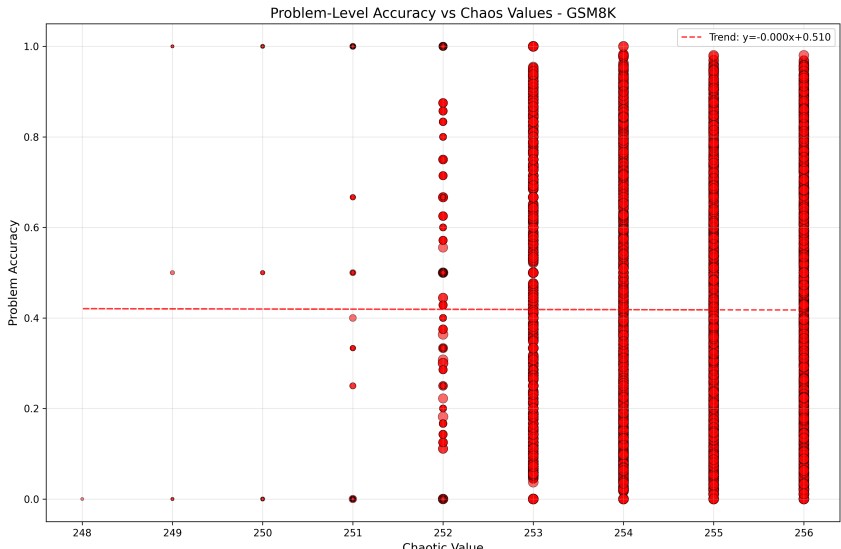

Figure S1: Correlation between generation order and accuracy. The x-axis shows how "chaotic" a generated sample is measured in the Hamming distance of its decoding order from a strict left-to-right (AR) order. The y-axis is the average accuracy of the generated samples with the same chaotic value for a problem. The size of the point represents the number of samples. We find no correlation, indicating that decoding in an AR-like order is not a predictor of success.

chance, follow an autoregressive-like order. If this were true, it would imply that the AR order is a fundamentally superior decoding path for the model.

We test this hypothesis directly with a large-scale correlation study. For a given dataset, we generate 256 samples for each problem using random remasking. For each generated sample, we compute two metrics:

**Accuracy:** A binary indicator of whether the final solution is correct.

**AR Similarity:** The Hamming distance between the sequence of positions unmasked in this sample and a canonical left-to-right (AR) order. A low Hamming distance indicates a generation order that is highly similar to AR.

If the hypothesis were correct, we would observe a strong negative correlation between the Hamming distance (difference from AR order) and accuracy; samples that decode in an AR-like order would be more likely to be correct.

Figure S1 plots these results, aggregating data across all samples of the same AR similarity for each problem in a dataset. The result is clear: we find no evidence of a correlation. The coefficient of the fitted lines for each dataset falls under the the order of magnitude of $10^{-3}$. This result holds consistently across all datasets and models we tested.

This null result is profound. It indicates that the autoregressive order is not a uniquely privileged path to a correct solution. Instead, the DLM has learned a rich, multi-faceted solution space where a correct answer can be reached through a vast plurality of different reasoning trajectories. The high *pass@k* achieved by random remasking is not due to it occasionally stumbling upon an AR order; it is due to the model's inherent ability to solve problems correctly via many diverse sequences of thought. The AR order's high *pass@k* is simply one manifestation of this general capability, not its source.

### A.3 Comparison Study of Greedy Decoding and Order-Token Search under Identical Temperature

To further validate the effectiveness of our approach, we compare greedy decoding with Order-Token Search under identical temperature settings. This controlled setup rules out confounding factors and highlights the contribution of the search strategy itself. As reported in Table S1, Order-Token Search delivers a remarkable performance boost on the Countdown dataset, confirming that the method provides tangible gains beyond simple greedy decoding.

Table S1: Countdown task performance under different configurations.

| Seq Len($L$), Diffusion Steps($S$), Beam Size($K$), Temperature($T$) | Accuracy (%) |
| --- | --- |
| L=128,S=64,K=1,T=0.0 | 20.7 |
| L=128,S=64,K=1,T=0.4 | 22.7 |
| L=128,S=64,K=5,T=0.4 | 34.4 |

### A.4 Additional Experimental Details

We provide further details on the experimental settings that complement the main results.

#### A.4.1 Beam Search Settings

Our **Order-Token Search** results (as shown in Table 1 and Table 2 is configured with beam sizes of $K \in \{3, 5, 8\}$ and block size of 32 along with a small search for the Gumbel noise temperature $\tau \in [0.2, 1.0]$, keeping in mind its role in balancing diversity and stability. As a general principle, a higher temperature introduces more diversity among the beams, but it can also risk destabilizing the token selection and decoding order. The settings used for our main experiments were chosen to maintain a reasonable balance between these factors.

In the main paper, we adopt the low-confidence remasking strategy together with the setting `gen_len = 2 × diffusion_steps;block_size = 32` for our baseline experiments. This configuration follows prior work (Zhao et al., 2025) and provides what can be regarded as a form of optimality: while it does not guarantee strict global optimality, it has been shown to yield a reasonably effective and competitive baseline under low-confidence conditions. Random-remasking majority-voting and Order-Token Search both use the same configuration. And we change `block_size = gen_len = 1` to simulate AR decoding on AR, AR majority-voting and AR Beam Search.

For the Order Search and Token Search experiments reported in Table 2, we use the configuration with $K = 3$ and $\tau = 0.0$. For Table 3, we adopt a setting of `gen_len = 512` and $\tau = 0.7$. The two sets of Countdown accuracies are obtained under different configurations and are serving different purposes. In the main results table, we report benchmark-level performance: we examine different generation lengths and report with the optimal temperature. By contrast, Table 3 is a controlled ablation where we fix the generation length to 512 and use a single temperature of $\tau = 0.7$, then vary only the beam size K to study how performance scales with K.

#### A.4.2 Pass@k Evaluation Settings

For pass@$k$ evaluation, we adopt the same configuration as in Yue et al. (2025). We set the temperature to $0.8$, which provides a balance between token diversity and plausibility. We use `gen_len = block_size = 256`, since the models we adopt are trained to generate sequences in a fully flexible order and we employ the same setup at inference time. For autoregressive decoding, we implement it via block diffusion with `block_size = 1`.

#### A.4.3 Computational Cost Analysis

Low-temp Order Search generally search only on the decoding order of sequences based on the confidence of each position. This algorithm is designed based on the intuition that decoding order might change the ultimate accuracy. Therefore, at every step, we keep $K$ positions that have the

highest probability from model logits independently unmasked. We then perform a look-ahead at the next step to have $K^2$ candidate sequences each with one more position unmasked. We calculate the confidence score and keep the top-$K$ candidates. In the experiment, we adopt the configuration of $K = 3, T = 0.0$, gen_len $= 256$ and the results also witness promising improvement.

However, the algorithm is computationally expensive, for it requires $K^2 \times$ gen_len forward passes in total. With $K = 3$ and gen_len $= 256$, this amounts to $3^2 \times 256 = 2304$ forward evaluations. In contrast, our Order-Token Search with $K = 5$ requires only $(128 \times 5) + \left(\frac{128}{32}\right) \times 25 = 740$ forward passes, where $128/32$ corresponds to the number of blocks and each block update involves $5 \times 5$ expansions.

Low-temp Token Search is closely related to Order Search, but instead of expanding $K$ positions at each step, it expands the top-$K$ most confident tokens for a single position. Starting from $K$ sequences, this again produces $K^2$ candidate sequences per step, leading to the same overall complexity of $K^2 \times$ gen_len forward passes. For instance, with $K = 3$ and gen_len $= 256$, Token Search also requires $3^2 \times 256 = 2304$ forward evaluations. Although the search space differs (token values vs. decoding order), the computational burden remains quadratic in $K$, making it substantially more expensive than our Order-Token Search, which scales only linearly with $K$.

## A.5 COMPUTATION COMPLEXITY

To manage computational complexity, we deliberately structure the search using **block diffusion** (Arriola et al., 2025; Nie et al., 2025). This avoids the prohibitive cost of a naive search at every step, which would incur a complexity of $O(S \cdot K^2 \cdot L)$, where $S$ is the number of diffusion steps. The overhead of our approach is far lower. Let $L$ be the generation length, $K$ the beam size, and $B$ the number of blocks. The total Number of Function Evaluations (NFE) for OTS is the sum of $K$ independent denoising trajectories (costing $S \cdot K \cdot L$) and the likelihood evaluations for search, which are performed $B$ times at block boundaries (costing $B \cdot K^2 \cdot L$).

Thus, the total NFE is NFE(OTS) $\approx S \cdot K \cdot L + B \cdot K^2 \cdot L$. In our main experimental setting (where $S = L/2$ and $B = L/32$), with a typical beam size $K \approx 4$, this simplifies to NFE(OTS) $\approx (L^2 \cdot K)/2 + (K^2 \cdot L^2)/32 \approx 2.5 \cdot L^2$. This is critically important, as it is directly comparable to the NFE of a standard majority-voting baseline with 5 samples: NFE(MV-5) $= S \cdot 5 \cdot L = (L/2) \cdot 5 \cdot L = 2.5 \cdot L^2$. Therefore, OTS provides a structured joint search over the (order $\times$ token) space at roughly the same computational cost as a widely-used unstructured sampling baseline. After processing all blocks, the single best sequence is selected from the final $K$ candidates based on the highest likelihood. To further validate our analysis, we measured wall-clock time on the Countdown dataset, averaging over all problems and comparing low-confidence remasking, naive majority voting (5 samples), and OTS with 4 beams. As shown in Table S2, our optimized implementation of OTS runs faster than the majority-voting baseline.

Table S2: Wall-clock time (in seconds) comparison on the Countdown dataset, averaged over all problems. This demonstrates that OTS (4 beams) is roughly 2-3x slower than a single low-confidence run, but about 2x faster than majority voting with 5 samples.

| Method / Generation length | 64 | 128 | 256 | 512 |
| --- | --- | --- | --- | --- |
| Low-confidence remasking | 1.55 | 3.19 | 6.60 | 14.52 |
| + Majority-voting (5 samples) | 7.73 | 15.94 | 32.99 | 72.59 |
| Order-Token Search (4 beams) | 3.52 | 7.46 | 16.64 | 40.41 |

## A.6 ORDER-TOKEN SEARCH SCALING WITH NFE

In this section, we analyze how OTS scales with test-time compute on the Countdown benchmark, comparing it to majority-voting strategies under roughly matched FLOP budgets. Figure S2 plots accuracy as a function of NFE by varying the beam size for OTS and the number of samples for AR+MV and Random+MV. At the matched-compute frontier, OTS with beam size 6 achieves 29.3% accuracy, while AR+MV and Random+MV peak at 19.9% and 18.4%, respectively. Moreover, OTS continues to gain accuracy as beams are added (from 16.0% at beam 1 to 29.3% at beam 6), whereas majority-voting baselines only exhibit marginal returns as more samples are drawn. This dominance

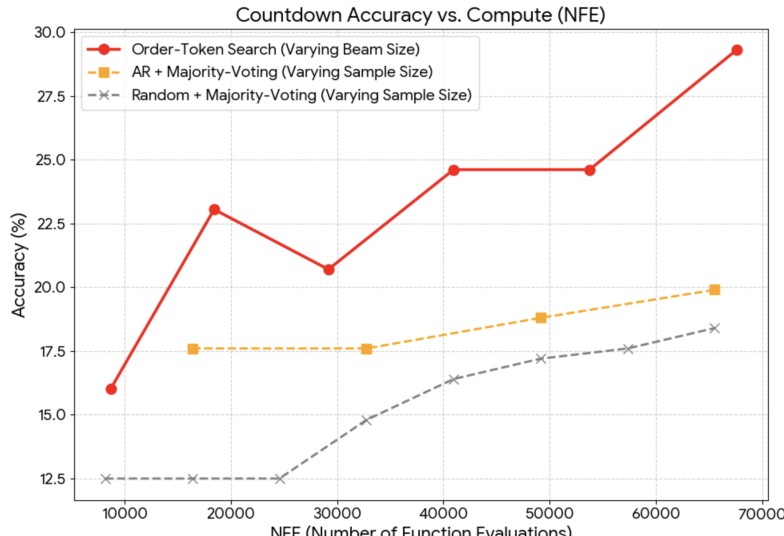

Figure S2: Countdown accuracy versus test-time compute (NFE) for OTS and majority-voting baselines. For each method, we vary beam size (OTS) or the number of samples (AR+MV, Random+MV), and choose the largest configuration so that all right-most points have roughly matched NFE. At this matched-compute point, OTS with beam size 6 attains 29.3% accuracy, compared to 19.9% for AR+MV and 18.4% for Random+MV, indicating more efficient use of additional FLOPs than simply drawing more independent diffusion samples.

in the accuracy–NFE plane shows that jointly searching over orders and tokens turns extra compute into substantially larger performance gains than standard multi-sample diffusion decoding.

## A.7 SENSITIVITY OF ORDER-TOKEN SEARCH TO BLOCK SIZE

Our scoring function $s(x_t; x_s)$ is explicitly designed to be stable across a range of block sizes. Conceptually, the block size controls a bias–variance trade-off in likelihood estimation. When the block is larger, the model must jointly predict more tokens at once, making each scoring step harder but fewer in number. When the block is smaller, each prediction is easier and closer to the MDM training distribution—where the model typically denoises a limited number of masks at a time—but search is invoked more frequently. In all cases, the score of a candidate is the sum of these incremental block-level log-likelihoods over its full generation path (Eq. 2), so changing the block size simply changes how finely this path-wise likelihood is decomposed, not the underlying distribution being estimated. We therefore view the block size primarily as an efficiency and granularity knob rather than a fragile hyperparameter for the scoring rule itself.

In practice, we find that Order-Token Search is not highly sensitive to the exact block size within a reasonable range. On MATH500 with generation length 128, sweeping the block size from 1 to 128 yields accuracies between 23.0% and 28.0%. Across block sizes 2–64, performance stays in a narrow band around 26.5% (approximately 26.5±1.5), and all such settings significantly outperform the degenerate cases of block size 1 and 128, where Order-Token Search either loses the order space entirely (block size 1) or forces the model to effectively denoise the entire sequence in one shot (block size 128). This empirical plateau for intermediate block sizes matches the bias–variance trade-off discussed above and supports the view that block size primarily controls the efficiency and granularity of search rather than acting as a delicate tuning parameter. The full sweep is visualized in Figure S3.

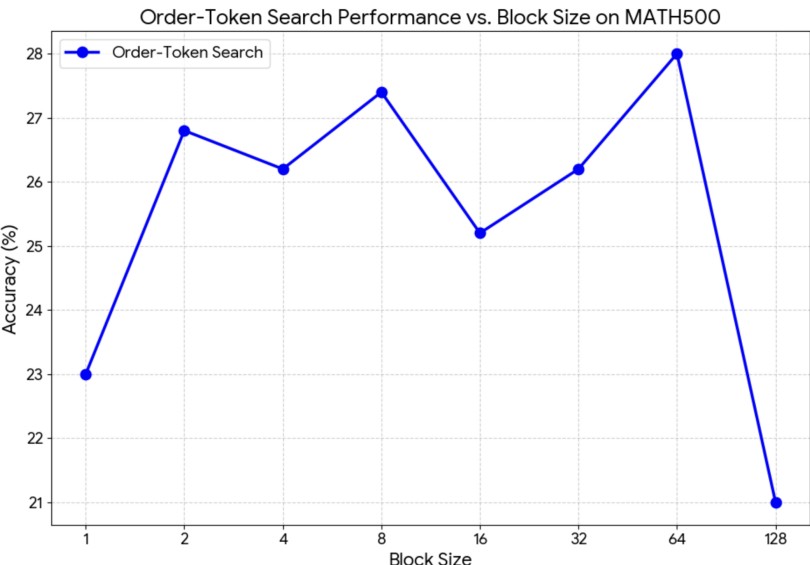

Figure S3: **Effect of block size on OTS accuracy** on MATH500 with generation length 128. Accuracy remains stable for block sizes 2–64, while the degenerate settings of block size 1 and 128 significantly underperform, confirming that block size mainly acts as an efficiency and granularity knob.

