# OpenReview forum: "Improving Diffusion Language Model Reasoning through Joint Search in Generation Order and Token Space"
_ICLR.cc/2026/Conference — Submitted to ICLR 2026_

### Official Review · Reviewer_HT8y · 2025-10-27

**Soundness:** 2
**Presentation:** 3
**Contribution:** 1
**Rating:** 2
**Confidence:** 4

**Summary:**

This paper describes a new method for remasking in the denoising process of Diffusion Language Models that searches jointly over generation orders and token choices to maintain high accuracy while promoting diverse reasoning. The paper empirically validates their proposed method in comparison with standard remasking strategies over various mathematical reasoning and problem solving benchmark datasets.

**Strengths:**

- The paper is very well written
- The problem setting is clearly stated and motivated
- Experimental results are clearly presented

**Weaknesses:**

- For a paper with only experimental results, the extent of the experiments seem lacking to prove significance. It seems that only the Countdown task exhibits strong performance gains and little explanation is given for why that seems to be.
- The computational complexity of the Order-Token Search is not clearly stated, is there a strong incentive to use this new algorithm despite marginal gains in performance?
- What is being meant by “reasoning” in this paper? What is it about the new masking strategy that unlocks “reasoning” capability in the MDMs, which seems to be a central claim of the study.
- The case study is informative to the intuition behind the proposed algorithm, but one example is not convincing to the soundness of an algorithm. Are there other wider trends or more concrete theoretical explanations that hint at better performance on the benchmark datasets?

**Questions:**

See Weakness section.

---

> ### Author Response · Authors · 2025-11-20
>
> We appreciate the reviewer’s recognition that our paper “is very well written” and, most importantly, validates Order-Token Search “in comparison with standard remasking strategies over various mathematical reasoning and problem solving benchmark datasets.”
>
> > **W1: “What is being meant by “reasoning” in this paper? What is it about the new masking strategy that unlocks “reasoning” capability in the MDMs?”**
>
> We thank the reviewer for asking what we mean by “reasoning” in the context of diffusion language models. For a high-level description of our reasoning formulation and the Order–Token Search (OTS) algorithm, please see General Response 2; here we focus on clarifying the specific points raised in this question. In short, we adopt a task-level view: each problem induces a latent graph of logical dependencies, and a DLM decoding trace corresponds to a trajectory through this graph, formed by repeatedly choosing **where** in the sequence to denoise and **what** token to write there.
>
> Standard MDM training only directly supervises token predictions under random remasking, so the induced distribution over trajectories is determined implicitly by the inference-time remasking rule. Low-confidence remasking hard-codes a single heuristic traversal, which improves pass@1 but restricts exploration of alternative orders that might better respect the underlying dependency structure. OTS is designed to address precisely this limitation: instead of committing to one low-confidence trajectory, it uses the model’s parallel denoising capability to expand and score multiple candidate sequences, thereby exploring different combinations of positions and intermediate steps before selecting the highest-scoring trajectory.
>
> Thus, the “reasoning capability” does not come from a “new masking strategy”, but from enabling the model to search over multiple plausible denoising orders and token assignments, rather than being locked into a single heuristic remasking strategy. This explicit exploration of alternative trajectories is what allows OTS to uncover higher-quality solutions on multi-step math and logic benchmarks.
>
> > **W2: “The extent of the experiments seem lacking to prove significance.” Why “Countdown task exhibits strong performance gains”?**
>
> We appreciate the reviewer’s concern about the breadth of the experimental evaluation. We would like to emphasize that across four benchmarks, OTS matches or surpasses the performance gains of post-training methods such as diffu-GRPO on three of them, while using orders of magnitude less compute. Achieving post-training-level improvements at inference time alone is a strong indication that OTS exploits a genuine modeling opportunity in diffusion language models, not just a heuristic remasking trick. This observation supports our core claim: DLMs inherently define a joint generative process over token choices and decoding orders, yet training only supervises the former. OTS restores the ability to explore multiple orders, which is especially critical for multi-step reasoning tasks.
>
> The Countdown task naturally amplifies this effect. Countdown requires constructing a composition of arithmetic operations from a given set of numbers to reach a target; success hinges not only on selecting good intermediate operations (tokens) but also on choosing a productive sequence of partial computations (order). Because the sequence in which intermediate steps are fixed can determine whether the final objective is even reachable, explicitly exploring alternative order paths—something OTS uniquely enables—yields particularly large gains on this dataset. In this sense, Countdown is not an outlier but a clean, high-signal instance of our broader point: when solving a problem requires discovering both what intermediate steps to take and in which order to take them, DLMs benefit substantially from order-aware decoding.
>
> To further strengthen our empirical case, we have expanded our comparison to include a richer set of baselines and methods—AR, AR with majority voting, low-confidence + majority voting, as well as a stronger backbone model (LLaDA-1.5). Across these additions (see Table R1 and R2 in General Response 1), the pattern remains consistent: enabling order exploration via OTS yields material and robust improvements in DLM reasoning performance. For example, with the LLaDA-1.5 model, OTS improves the overall accuracy across four benchmarks (the “All” metric) from the second best baseline 35.0 (Low-conf + MV) to 36.7. On MATH500, OTS attains the best average accuracy of 33.8, outperforming the strongest non-OTS baselines such as Low-conf + MV (31.3) and AR + beam search (29.9). On Countdown, OTS achieves an average of 28.0, again clearly surpassing Low-conf + MV (22.5) and AR + beam search (18.1). We hope these expanded results and clarifications address the reviewer’s concern that improvements were confined to a single task.

---

> > ### Author Response · Authors · 2025-11-20
> >
> > > **W3: “The computational complexity of the Order-Token Search is not clearly stated, is there a strong incentive to use this new algorithm despite marginal gains in performance?”**
> >
> > We thank the reviewer for raising the question of computational complexity and practical incentive. First, we clarify that the performance gains achieved by OTS are not marginal: across three out of four reasoning benchmarks, OTS matches or surpasses the improvements obtained by heavy post-training methods such as diffu-GRPO, while requiring no additional model training. This level of improvement at inference time alone represents a strong incentive, particularly for settings where post-training is infeasible due to compute or data constraints. Furthermore, OTS consistently outperforms comparable-compute inference-time methods such as majority voting, demonstrating that its improvements cannot be attributed merely to extra sampling.
> >
> > On computational cost, we apologize that Section 4.1 did not fully expand on the complexity analysis. We provide it here for completeness. Let S be the number of diffusion steps, L the sequence length, K the beam size, and B the number of blocks. Vanilla DLM inference requires O(S × L) model evaluations. A naive form of OTS, which performs order search at every step, would cost O(S × K² × L) due to keeping K independent beams and rolling out K candidate samples for each beam.
> >
> > However, our method is explicitly designed to avoid this quadratic overhead. OTS only performs search at block boundaries, reducing the search cost to B × K² × L additional forward passes. The full NFE of OTS is therefore approximately S × K × L (from K parallel denoising trajectories) plus B × K² × L (from likelihood evaluation during search). In our main experiments, where S = L/2 and B = L/32, this becomes NFE(OTS) ≈ L² × K/2 + K² × L²/32. With our typical choice of K ≈ 4, this simplifies to roughly 2.5 × L², which is comparable to the compute cost of majority voting with five samples (NFE ≈ 2.5 × L²).
> >
> > Thus, OTS provides post-training-level improvements at roughly the same compute as majority-voting baselines, offering a strong practical incentive with no prohibitive overhead. We believe this strikes an attractive balance between performance and efficiency.
> >
> > > **W4: “The case study is informative to the intuition behind the proposed algorithm, but one example is not convincing to the soundness of an algorithm.”**
> >
> > We appreciate the reviewer’s concern about relying on a single case study. While the Countdown example helps build intuition, the broader evidence across our experiments shows that OTS’s improvements are systematic rather than anecdotal. Across four benchmarks and multiple sequence lengths, OTS consistently outperforms both low-confidence remasking and random-remasking–based strategies, and in three out of four benchmarks it reaches or surpasses the gains of post-training (diffu-GRPO) despite requiring only inference-time computation. These cross-task trends reinforce our central claim: diffusion language models implicitly define a joint generative space over token choices and decoding orders, yet existing decoding methods do not exploit both sides sufficiently. OTS performs structured order-token exploration, allowing the model to recover globally coherent reasoning chains that greedy / order-agnostic decoding frequently miss.
> >
> > Beyond main-table accuracy, we also observe consistent patterns that further validate the soundness of OTS. After studying the effect of temperature on low-confidence remasking, we find that OTS not only outperforms its low-confidence counterpart at every temperature, but that the performance margin grows as temperature increases, indicating that default decoding lacks exploration while OTS actively unlocks additional reasoning paths. For example on GSM8K, OTS beats low-confidence remasking by +2.0 points at T=0.2, +4.1 at T=0.5, +5.2 at T=0.8, and +6.0 at T=1.0. This trend provides further evidence that the improvements are not tied to a single curated example: whenever reasoning requires navigating multiple plausible solution orders—as is common in arithmetic, algebraic, and multi-step logical tasks—OTS’s joint order-token search reliably yields better outcomes than purely token-based heuristics.

---

### Official Review · Reviewer_TD2K · 2025-10-31

**Soundness:** 1
**Presentation:** 3
**Contribution:** 2
**Rating:** 2
**Confidence:** 4

**Summary:**

This paper proposes a search strategy for generation with discrete diffusion models to enable both high quality and exploration compared to things like low-confidence remasking which sacrifice exploration for quality. At a high-level it is a kind of beam search over both token orderings and token selections. Specifically they adopt the block-wise autoregressive decoding pattern commonly used in past work. They generate multiple possible blocks and only keep the candidates with the highest scores, similar to beam search. For their proposed scoring function, they unmask all tokens and compute the likelihood of the (re-masked) block conditioned on all other tokens. They report results across different decoding methods for standard math benchmarks and some synthetic puzzle benchmarks.

**Strengths:**

One of the most interesting aspects of this class of generative models is their improved decoding flexibility compared to autoregressive models. Exploring decoding strategies over both positions and tokens that are only possible for this class of models is an interesting research direction.

The motivation from the limitation of existing approaches (i.e. random is diverse but low-quality, and high-confidence is high-quality but not diverse) is clear and intuitive.

A naive beam search over every decoding choice would be extremely expensive. Performing it block-wise is a clever way to balance the benefits of search while reducing the computational overhead.

Their method outperforms low-confidence sampling (see question 2) and random sampling with majority baseline. Their ablation is Table 2 demonstrates the benefits of their joint search.

**Weaknesses:**

Although their approach is motivated by balancing exploration and quality, this benefit is not validated for their method. They present Pass@k curves to argue for the limitation of low-confidence remasking strategies, but never present such a curve for their approach to validate that it solves the problem with low-confidence remasking. For autoregressive models, beam search often has diversity issues, so a similar thing could be happening here.

Although the asymptotics of their search algorithm is discussed, the computational cost (at least in terms of NFE) of various decoding settings should be reported alongside all results. There are many ways to improve performance by expending more compute. It is important to quantify this rigorously to ensure that the method does not impose unreasonable tradeoffs. In general for test-time scaling approaches like these, results should be reported across a range of inference-matched settings to get a true picture of the tradeoffs.

Some additional decoding baselines should be included. Based on their own figure 2, autoregressive decoding appears to be a very strong baseline and should be included. Majority voting with both random and autoregressive baselines in compute matched settings should be included.

In the appendix, the authors mention that they add gumbel noise to the logits to improve exploration. This is not discussed in the main paper. This feels like an important implementation detail that should not be relegated to the appendix. Is this critical? How does Order-Token Search perform without it? Can you similarly apply this perturbation to low-confidence re-masking to improve diversity. A majority voting baseline with low-confidence remasking and gumbel noise should also be reported, given the use of gumbel noise in the proposed method.

The analysis in A.2. is not convincing. Figure 2 shows that restricting the model to the autoregressive ordering is extremely effective, dramatically outperforming the other decoding strategies in most settings. Analyzing whether correct solutions under random sampling arose from landing on the autoregressive ordering is a very indirect way to study it. The odds of actually achieving the autoregressive ordering (or even getting very close to it) when doing random decoding is extremely small as demonstrated by the overwhelming amount of large chaotic values. The finding in Figure 2 pretty strongly suggests that the autoregressive ordering is particularly effective, at least for many tasks and settings.

**Questions:**

1. Equation two doesn’t seem to batch the text and figure. My understanding is that you are measuring the likelihood of the block given the surrounding context. Is that correct? As written, it is the probability of the clean sequence conditioned on the block.
2. What is the performance of the proposed method without Gumbel noise?
3. What are the total NFEs for all reported evaluation settings?

---

> ### Author Response · Authors · 2025-11-20
>
> We appreciate the reviewer’s recognition that the motivation of our work “from the limitation of existing approaches is clear and intuitive,” as well as that performing Order-Token Search “block-wise is a clever way to balance the benefits of search while reducing the computational overhead.”
>
> > **W1: “Never present [a pass@k] curve for their approach to validate that it solves the [low-pass@k] problem with low-confidence remasking.”**
>
> We thank the reviewer for this thoughtful comment. Our analysis of pass@k in Section 3 is intended as a diagnostic showing the exploration–quality trade-off of existing decoders, not as the main evaluation target for our method. As we state at the end of Section 3, this analysis “reveals the core opportunity in DLM decoding: achieving higher accuracy than low-confidence remasking requires strategic exploration of both generation order and token space” and motivates “an algorithm that actively searches this joint space within a single decoding process to locate correct answers that greedy strategies miss.” In other words, OTS is specifically designed to **turn the model’s multi-sample potential (high pass@k of diverse strategies) into higher pass@1**. That is why our main tables focus on accuracy (pass@1): the question we aim to answer is whether structured, internal exploration can beat low-confidence remasking on a single shot, which our results consistently confirm across four reasoning benchmarks and multiple backbones.
>
> More broadly, our central claim and contribution are summarized in General Response 2, where we formalize DLM decoding as trajectory selection over both generation orders and token choices and position OTS as a structured search procedure over this space. In the context of this comment, the key point is that OTS is not optimized to maximize pass@k directly, but to convert the model’s multi-sample potential into a more capable single-sample reasoning decoder—one that jointly decides what to say and in what order—thereby yielding higher pass@1 than existing low-confidence remasking strategies.
>
> > **W2: “The computational cost (at least in terms of NFE) of various decoding settings should be reported alongside all results.” “Results should be reported across a range of inference-matched settings to get a true picture of the tradeoffs.”**
>
> We thank the reviewer for emphasizing the importance of compute-matched comparisons. We agree that test-time scaling methods must be evaluated under rigorously controlled budgets. In our setting, the dominant cost is the number of model forward calls, which we track as the number of function evaluations (NFE). Let S be the number of diffusion steps, L the generation length, K the beam size, and B the number of blocks. Vanilla DLM decoding has complexity O(S·L). A naive version of OTS that searches at every step would cost O(S·K²·L), since we maintain K beams and roll out K candidates per beam. Our actual algorithm deliberately avoids this: we only perform search at block boundaries, so NFE(OTS) ≈ S·K·L (K independent denoising trajectories) + B·K²·L (likelihood evaluations for search). In the main configuration, where S = L/2 and B = L/32, this simplifies to NFE(OTS) = L²·K/2 + K²·L²/32. With our typical choice K ≈ 4, this yields NFE(OTS) ≈ 2.5·L², which is comparable to the NFE of a standard majority-voting baseline with 5 samples under the same diffusion schedule: NFE(MV-5) = S·5·L = 2.5·L². Thus, OTS improves performance primarily through better use of the same test-time compute as a widely accepted sampling strategy, rather than simply spending more FLOPs.
>
> In Table 1, all comparisons between OTS and majority-voting baselines are made under this matched setting. The table also varies L across multiple values (e.g., 64, 128, 256, 512), which already provides a test-time scaling view: as inference budget increases with L, OTS consistently delivers higher accuracy than baselines at similar NFE. In the revised manuscript (Appendix A.5), we have made this explicit by (i) reporting NFE alongside each method and (ii) highlighting that OTS and majority voting are compute-matched across all reported configurations, so that the reported gains cannot be attributed to unbounded extra test-time compute.

---

> > ### Author Response · Authors · 2025-11-20
> >
> > > **W3: “Some additional decoding baselines should be included.” AR and majority-voting.**
> >
> > We thank the reviewer for these helpful suggestions and agree that including stronger decoding baselines strengthens our claims. Our original experiments already evaluated low-confidence remasking, random remasking with majority voting. Following the reviewer’s feedback, we have now explicitly added (i) AR decoding and (ii) majority voting with AR samples as additional baselines in our main results. Together, we now cover majority voting for both random-remasking samples and AR-order samples, alongside low-confidence remasking and OTS. As discussed in our response to W2, we ensure these new baselines are compute-matched with OTS by aligning the number of function evaluations (NFE). In Table R1 and R2 (General Response 1), we observe that OTS continues to outperform all decoding baselines on most benchmarks while using comparable test-time compute (e.g., for LLaDA-1.5 on Countdown AR: 14.4, AR + MV-5: 15.1, OTS: 28.0; on MATH500 AR: 26.7, AR + MV-5: 29.1, OTS: 33.8).
> >
> > > **W4: “Add gumbel noise to the logits to improve exploration … is not discussed in the main paper.” “How does Order-Token Search perform without it?” “A majority voting baseline with low-confidence remasking and gumbel noise should also be reported.”**
> >
> > We thank the reviewer for raising this concern. Our inference stack follows the standard masked-diffusion setup (e.g., d1 [1]), where adding Gumbel noise to the logits is simply the mechanism used to implement temperature-controlled categorical sampling. Gumbel noise is not a special ingredient unique to OTS, but just how “temperature > 0” sampling is realized for all stochastic methods we evaluate.
> >
> > Conceptually, OTS does require some exploration to be effective: if we remove sampling noise entirely (temperature = 0), OTS collapses to the deterministic low-confidence baseline, so it does not offer additional benefits in that fully-greedy regime. To verify that OTS is not merely benefiting from extra randomness, we ran a temperature sweep where both low-confidence remasking and OTS use the same non-zero temperatures and Gumbel-based sampling. On GSM8K with generation_length = 128, OTS consistently outperforms low-confidence remasking at every tested temperature and the gap widens as exploration increases:
> >
> > | Temperature | Low-confidence | OTS   | Margin |
> > |-------------|----------------|-------|--------|
> > | T = 0.2     | 69.98          | 71.95 | +2.0   |
> > | T = 0.5     | 70.36          | 74.45 | +4.1   |
> > | T = 0.8     | 68.16          | 73.31 | +5.2   |
> > | T = 1.0     | 67.17          | 73.16 | +6.0   |
> >
> > This shows that, when both methods are given the same Gumbel-based perturbations, OTS consistently turns that exploration into higher accuracy; its benefit comes from structured joint search over order and tokens, not from a hidden source of randomness.
> >
> > Following the reviewer’s suggestion, we also introduce a majority-voting baseline that combines low-confidence remasking with Gumbel noise (i.e., temperature-based sampling) and report it under compute-matched settings to OTS (same NFE). As shown in Table R1 and R2 (General Response 1), even when both methods use the same Gumbel-based sampling and are matched for compute, OTS remains clearly stronger. For example, on MATH500, OTS improves over Low-conf+MV from 29.7→32.8 (+3.1) for LLaDA and 31.3→33.8 (+2.5) for LLaDA-1.5 (averaged over lengths). On Countdown, the gains are even larger: from 20.7→28.4 (+7.7) for LLaDA and 22.5→28.0 (+5.5) for LLaDA-1.5. Aggregated over 4 benchmarks (“All”), OTS improves accuracy from 32.5→35.2 (+2.7) and 35.0→36.7 (+1.7), respectively.
> >
> > We hope these results address the reviewer’s concern: we now (i) explained Gumbel sampling, (ii) show that OTS outperforms low-confidence remasking under the same temperature and sampling scheme, and (iii) report a Gumbel-based majority-voting baseline that is compute-matched yet still weaker than OTS.
> >
> > [1] Zhao et al., arXiv 2025. d1: Scaling reasoning in diffusion large language models via reinforcement learning.

---

> > > ### Author Response · Authors · 2025-11-20
> > >
> > > > **W5: In Appendix A.2 “analyzing whether correct solutions under random sampling arose from landing on the autoregressive ordering is a very indirect way to study it.”**
> > >
> > > We appreciate the reviewer’s careful reading of Appendix A.2 and Figure 2, and we agree that our current analysis of autoregressiveness is preliminary. The goal of A.2 was not to claim that autoregressive (AR) ordering is ineffective—in fact, Figure 2 clearly shows that AR-style decoding is a very strong baseline in terms of pass@k—but rather to ask a narrower, exploratory question: among the correct solutions obtained under random DLM decoding, do we see a strong tendency for those trajectories to align closely with the AR order? Our “chaos” statistic was an indirect proxy to probe this, and as the reviewer correctly notes, random decoding almost never lands exactly on the AR order, which makes this kind of correlation analysis inherently noisy. Because of these limitations and its exploratory nature, we relegated A.2 to the appendix and do not base any of our main claims on it.
> > >
> > > More broadly, we fully agree that understanding the special role (if any) of the AR order for diffusion language models is an interesting open question, and our work is not intended to argue against its effectiveness. Our main goals and contributions—framing DLM decoding as trajectory selection and introducing Order–Token Search as a structured search procedure—are summarized in General Response 2 and in the introduction of the revised manuscript. In this light, Appendix A.2 should be read as a preliminary, exploratory analysis of how AR-like orders appear under random decoding, not as a central claim of the paper. A more rigorous, dedicated study of how AR-like orders interact with DLM training and decoding is an exciting direction that we view as complementary to our contribution, and we plan to pursue it in future work.
> > >
> > > > **Q1: “Equation two doesn’t seem to batch the text and figure.” “As written, it is the probability of the clean sequence conditioned on the block.”**
> > >
> > > The reviewer is correct that we measure the likelihood of the blocks prescribed by function *b* given the surrounding context. Equation 2, `p(x_0 | b(x_s, x_t, x_0))`, precisely reflects this operation. The function *b* identifies the specific blocks of token positions `{ i | x_t,i = MASK and x_s,i != MASK }` that were denoised between time *t* and *s*, masks these blocks in `x_0`, and returns the masked sequence. Now `p(x_0 | partially masked x_0)` gives the likelihood of the masked block. We have included this elaboration in the revised Section 4.2.
> > >
> > > > **Q2: “What is the performance of the proposed method without Gumbel noise?”**
> > >
> > > We kindly refer the reviewer to our response to W4.
> > >
> > > > **Q3: “What are the total NFEs for all reported evaluation settings?”**
> > >
> > > We kindly refer the reviewer to our response to W2.

---

### Official Review · Reviewer_RQJW · 2025-11-01

**Soundness:** 2
**Presentation:** 3
**Contribution:** 3
**Rating:** 6
**Confidence:** 3

**Summary:**

This paper proposes Order-Token Search, a novel decoding algorithm for Diffusion Language Models that jointly explores generation orders and token choices to improve reasoning performance. The work addresses a trade-off in DLM decoding strategies provides improvements on reasoning-intensive tasks. The work is generally interesting and highlights the importance of masking order in generation, and introduces a new test-time scaling method for DLM.

**Strengths:**

1. Novel problem identification: The paper highlights current limitation in diffusion language model decoding, which lacks the generation diversity offered by AR. The systematic analysis of the pass@1 vs pass@k trade-off in existing decoding strategies (Section 3) is compelling, showing that low-confidence remasking improves single-trial accuracy but limits exploration diversity.

2. Improved Empirical Results: The method achieves consistent improvements across multiple benchmarks, with particularly notable gains on Countdown that rival post-training methods. The comparison showing Order-Token Search outperforming computationally expensive baselines like Order Search and Token Search demonstrates the value of the dedicated likelihood estimation.

**Weaknesses:**

1. The decoding method computation complexity is not shown empirically. How does the method perform compared to baselines under the same FLOPs, e.g. draw a test-time scaling curve?

2. Only one backbone model is studied. The paper would hold a stronger claim with more than one pretrained backbones studied.

3. Scoring Function Stability: How sensitive is the block-level likelihood estimation to the choice of block size?

Minor Issues

- The notation in Equation 2 could be clearer - the function b(xs, xt, x0) is introduced but not precisely defined.

**Questions:**

See weaknesses

---

> ### Author Response · Authors · 2025-11-20
>
> We appreciate the reviewer’s recognition that our work “highlights current limitation in diffusion language model decoding,” as well as that Order-Token Search “demonstrates the value of the dedicated likelihood estimation” and  achieves “particularly notable gains on Countdown that rival post-training methods.” We now address their remarks.
>
> > **W1: “The decoding method computation complexity is not shown empirically. How does the method perform compared to baselines under the same FLOPs, e.g. draw a test-time scaling curve?”**
>
> We thank the reviewer for highlighting the importance of compute when evaluating decoding methods. Below we provide a comprehensive analysis on the time complexity of Order-Token Search (OTS) and justify its performant advantage over the majority-voting baseline that shares the same FLOPs.
>
> Let S be the number of diffusion steps, L the generation length, K the beam size, and B the number of blocks. Vanilla DLM inference has time complexity O(S·L). A naive version of OTS that searches at every step would cost O(S·K²·L), since we maintain K beams and roll out K candidates per beam. Our actual implementation balances search and efficiency by triggering search only at block boundaries, leading to roughly S·K·L forward passes for denoising plus B·K²·L extra passes for likelihood evaluation. In the main table setting (S = L/2, B = L/32, K ≈ 4), this yields NFE(OTS) ≈ 2.5·L², which is comparable to the NFE of majority voting with 5 samples, NFE(MV-5) = S·5·L = 2.5·L². Since FLOPs scale linearly with the number of forward evaluations for a fixed model and sequence length, this gives a direct, apples-to-apples comparison of OTS and majority voting under approximately matched test-time compute.
>
> Per the reviewer’s suggestion, we have also added an empirical test-time scaling study. We fix temperature = 0.8 and generation_length = 128, and vary the beam size for OTS and the sample size for majority-voting baselines on Countdown:
>
> | Method \ Beams |1 | 2 | 3 | 4 | 5 | 6 |
> |------------|-------|-------|-------|-------|-------|-------|
> | OTS  | 16.0 | 23.1 | 20.7 | 24.6 | 24.6 | 29.3 |
>
> | Method (compute) \ Samples                 | 1    | 2    | 3    | 4    | 5    | 6    | 7    | 8    |
> |-------------------------------|------|------|------|------|------|------|------|------|
> | AR+MV (2x)             | 17.6 | 17.6 | 18.8 | 19.9 |    |   |   |   |
> | Random+MV (1x)         | 12.5 | 12.5 | 12.5 | 14.8 | 16.4 | 17.2 | 17.6 | 18.4 |
>
> In the revised manuscript, Figure S2 plots Countdown accuracy as a function of NFE for OTS (varying beam size) and the majority-voting baselines (varying the number of samples). For each method, the right-most point corresponds to its largest beam/sample configuration, and these configurations are chosen so that their NFE values are roughly matched (all around the same FLOP budget). At this matched-compute point, OTS with beam size 6 attains 29.3% accuracy, whereas AR+MV reaches 19.9%, and Random+MV 18.4%, showing a clear accuracy gain at comparable FLOPs. Moreover, as we move along the curves, OTS consistently makes more effective use of additional compute: accuracy improves from 16.0% (beam 1) to 24.6% (beams 4–5) and finally 29.3% (beam 6), while the majority-voting baselines only exhibit marginal improvement when more samples are added (e.g., Random+MV increases from 12.5% (3 samples) to 14.8% (4 samples) but finally achieving 18.4% (8 samples)). Overall, OTS dominates the majority-voting strategies in the accuracy–NFE plane, indicating that our search over both order and tokens converts extra FLOPs into substantive performance gains more efficiently than simply drawing more independent diffusion samples.

---

> > ### Author Response · Authors · 2025-11-20
> >
> > > **W2: “Only one backbone model is studied. The paper would hold a stronger claim with more than one pretrained backbones studied.”**
> >
> > We thank the reviewer for this suggestion and agree that demonstrating robustness across backbones strengthens the paper. While our method is decoding-side and architecture-agnostic—it only assumes access to a masked diffusion language model—we have now extended our experiments beyond the original LLaDA backbone. In particular, we evaluate Order-Token Search on LLaDA-1.5 [1], a post-trained checkpoint starting from LLaDA but further optimized with reinforcement learning. Across all four benchmarks, OTS again provides consistent gains over low-confidence remasking and other baselines, mirroring the trends observed on the original backbone (e.g., LLaDA-1.5 + OTS improves from 67.7 to 72.2 on GSM8K, from 28.8 to 33.8 on MATH500, and from 19.8 to 28.0 on Countdown). This supports our claim that OTS is not tailored to a single model, but rather improves a range of diffusion LMs, including both SFT-trained and RL-trained variants.
> >
> > We are also in the process of running experiments on Dream-7B [2], which is trained with a different backbone and originally evaluated in tuned inference settings (8-shot prompting, temperature and top-p). Due to time constraints, we may only be able to include the full Dream results in a subsequent revision, but we do not expect them to alter our main conclusions.
> >
> > > **W3: “How sensitive is the block-level likelihood estimation to the choice of block size?”**
> >
> > We thank the reviewer for this thoughtful question. Our scoring function `s(x_t; x_s)` is explicitly designed to be stable across a range of block sizes. Conceptually, block size controls a bias–variance trade-off in likelihood estimation. When the block is larger, the model must jointly predict more tokens at once, making each scoring step harder but fewer in number. When the block is smaller, each prediction is easier and closer to the MDM training distribution—where the model typically denoises a limited number of masks at a time—but search is invoked more frequently. In all cases, the score of a candidate is the sum of these incremental block-level log-likelihoods over its full generation path (Eq. 2), so changing the block size simply changes how finely this path-wise likelihood is decomposed, not the underlying distribution being estimated. This is why we frame block size primarily as an efficiency and granularity knob rather than a fragile hyperparameter for the scoring rule itself.
> >
> > In practice, we find that OTS is not highly sensitive to the exact block size within a reasonable range. On MATH500 with generation length 128, sweeping the block size from 1 to 128 yields the following accuracy (in %):
> >
> > | Block Size | 1    | 2    | 4    | 8    | 16   | 32   | 64   | 128  |
> > | ---------- | ---- | ---- | ---- | ---- | ---- | ---- | ---- | ---- |
> > | Accuracy | 23.0 | 26.8 | 26.2 | 27.4 | 25.2 | 26.2 | 28.0 | 21.0 |
> >
> > Across block sizes 2–64, OTS remains consistently strong: performance stays within a narrow band (26.5 ± 1.5) and all settings substantially outperform the degenerate case of block size 1 and 128, where OTS completely loses the order space (block size 1) or the model must effectively denoise the entire sequence in one shot (block size 128). This empirical plateau for block sizes 2–64 matches the bias–variance trade-off discussed above and supports our view that block size is an efficiency and granularity knob rather than a delicate tuning parameter for OTS. We have included a plot of this ablation table and a short discussion in the revised version (Appendix A.7) to make this robustness more explicit.
> >
> > > **Minor Issue 1: “The notation in Equation 2 could be clearer - the function b(xs, xt, x0) is introduced but not precisely defined.”**
> >
> > We thank the reviewer for pointing this out. We have defined the function *b* in the updated manuscript: “The function *b* identifies the specific blocks of token positions `{ i | x_{t,i} = MASK and x_{s,i} != MASK }` that were denoised between time *t* and *s*, masks these blocks in `x_0`, and returns the masked sequence.”
> >
> > [1] Zhu et al., arXiv 2025. LLaDA 1.5: Variance-Reduced Preference Optimization for Large Language Diffusion Models.
> >
> > [2] Ye et al., arXiv 2025. Dream 7B: Diffusion Large Language Models.

---

> ### Comment · Reviewer_RQJW · 2025-11-27
> **Thank you for your reponse**
>
> Thank you for your response and the additional experiments. The scaling experiment appears strong, and I appreciate the added evaluations on new backbones and baselines.
>
> I am leaning toward accepting this work and would like to see more discussion with the other reviewers before I update my rating.

---

> > ### Author Response · Authors · 2025-11-27
> >
> > We sincerely thank Reviewer RQJW for the thoughtful follow-up and for engaging with our additional experiments. We are glad that the test-time scaling analysis and the new backbone results helped clarify the strengths and generality of Order-Token Search. We’re happy to answer any further questions that may come up in the discussion.

---

### Official Review · Reviewer_34SD · 2025-11-01

**Soundness:** 2
**Presentation:** 3
**Contribution:** 2
**Rating:** 2
**Confidence:** 5

**Summary:**

This paper proposes Order-Token Search, a structured search algorithm that jointly explores both generation order and token space. The method employs a likelihood-based scoring function to evaluate block-level denoising actions and prune unstable paths efficiently. Experiments on mathematical reasoning and planning tasks demonstrate the gains, suggesting that structured search can meaningfully enhance reasoning in DLMs without additional training.

**Strengths:**

1. Introduces a beam-search-like decoding algorithm (Order-Token Search) tailored for diffusion LMs, demonstrating improvement on challenging reasoning tasks.
2. Provides a clear analysis of the trade-off between accuracy and exploration in standard decoding methods (e.g., low-confidence remasking, AR-order, random sampling).

**Weaknesses:**

1. The paper fails to discuss sampling diversity controls (e.g., temperature, top-k/top-p sampling used in Dream) that could also improve pass@k performance when using low-confidence remasking methods, as studied in prior work (e.g., DiffuCoder [1], showing higher temperature will lead to more diverse token / order exploration and having high pass@k results).

2. In Figure 2, the “AR-order decoding” baseline lacks a beam search counterpart, leading to an unfair comparison—especially since the “token search” baseline is guided by other selected positions, as in line 412, "Token search is guided by a sequence of greedily-decided positions (selected via low-confidence remasking)". This undermines the validity of the baseline.

3. There is an inconsistency in Countdown accuracy between Table 3 (16–21%) and the main results (25.4–34.4%), which should be clarified.

4. The paper does not mention the extra inference overhead introduced by the joint search algorithm, which is important for evaluating practicality.

[1] https://arxiv.org/abs/2506.20639

**Questions:**

1. In Figure 1, the legend contains a typo: “within a forwar”
2. How is Order-Token Search integrated with full diffusion generation (rather than block diffusion)? Is it applied only to per-block, and does it scale with the full diffusion length?

---

> ### Author Response · Authors · 2025-11-20
>
> We appreciate the reviewer’s recognition that our work “provides a clear analysis of the trade-off between accuracy and exploration in standard decoding methods,” as well as that Order-Token Search (OTS) suggests “structured search can meaningfully enhance reasoning in DLMs without additional training.” We now address their remarks.
>
> > **W1: “The paper fails to discuss sampling diversity controls that could also improve pass@k performance when using low-confidence remasking methods.”**
>
> We thank the reviewer for raising the issue of sampling diversity controls. We believe the concern is mainly about Figure 2, where low-confidence remasking might seem constrained by a low-diversity sampling setup. As clarified in Appendix A.4.2, **all pass@k experiments in Figure 2 already use temperature = 0.8** “to provide a balance between token diversity and plausibility,” and this setting is applied uniformly to all decoding methods we compare (low-confidence remasking, random remasking, and AR baselines). We chose this configuration by following Yue et al. [1], who adopt the same temperature when studying pass@k for autoregressive LLMs. In other words, we do control sampling diversity in a way that is consistent with prior work, and we do so symmetrically across methods, ensuring that our conclusions are not driven by an artificially low-diversity configuration for low-confidence remasking. Top-k/top-p sampling plays a role similar to temperature in trading off diversity and plausibility; in this paper, we deliberately hold this diversity control fixed across methods to enable a clean comparison of decoding strategies.
>
> More broadly, our goal is *not* to propose yet another sampling-diversity heuristic, but to understand and improve *reasoning* in diffusion language models. As summarized in General Response 2, we **view DLM decoding as selecting trajectories in a joint space of generation orders and token choices**, and we introduce OTS as a structured search procedure over this space. Diversity controls such as temperature or top-k/top-p operate *within* a fixed remasking rule, whereas OTS changes the remasking behavior itself by explicitly exploring alternative orders and token assignments and then selecting among them. In this sense, our contribution is complementary: OTS is compatible with standard diversity controls and remasking strategies, and in our experiments we keep these controls fixed across methods precisely to isolate and highlight the effect of the decoding strategy.
>
> [1] Yue et al., NeurIPS 2025. Does reinforcement learning really incentivize reasoning capacity in llms beyond the base model?
>
> > **W2: “‘AR-order decoding’ baseline lacks a beam search counterpart, leading to an unfair comparison—especially since the ‘token search’ baseline is guided by other selected positions.”**
>
> We thank the reviewer for raising this concern and are happy to clarify the role of “AR-order decoding” in Figure 2. The intention of this baseline is diagnostic, not competitive: it is included to show that, even under a fixed left-to-right (AR) order, increasing token-level diversity already raises pass@k, indicating that token-space exploration matters. To then separate the effect of order vs. token exploration in a fair way, our main comparisons rely on OTS and the “token search” baseline, which share the same order distribution: both sample positions using low-confidence remasking. In other words, token search samples from exactly the same order distribution as OTS but only searches over tokens at each position, whereas OTS jointly explores both which position to denoise and which tokens to place. The performance gap between token search and OTS therefore cannot be attributed to differences in positional guidance, but instead reflects the added value of joint order–token search over token-only search under the same order distribution.
>
> That said, we agree that an explicit “beam-search counterpart” for the AR-order decoding baseline can further strengthen the comparison. As detailed in General Response 1, we have added AR + beam-search as a new baseline (equivalently, OTS with block_size = 1, which removes order search and reduces to AR-order decoding with beam search). Under this expanded set of baselines, OTS remains the strongest and most consistent decoding strategy on the substantive reasoning benchmarks. For example, averaged over all sequence lengths, OTS attains 32.8% vs. 26.0%/31.0% (AR/AR+beam) on MATH500 and 28.4% vs. 12.7%/21.2% on Countdown for LLaDA, and 33.8% vs. 26.7%/29.9% (MATH500) and 28.0% vs. 14.4%/18.1% (Countdown) for LLaDA-1.5. These consistent gains over both AR and AR + beam-search support our claim that jointly searching over generation orders and token sequences provides benefits beyond token-only exploration under a fixed AR order.

---

> > ### Author Response · Authors · 2025-11-20
> >
> > > **W3: “There is an inconsistency between Table 3 and the main results.”**
> >
> > The two sets of Countdown accuracies are obtained under different configurations and are serving different purposes. In the main results table, we report benchmark-level performance: we examine different generation lengths and report with the optimal temperature. By contrast, Table 3 is a controlled ablation where we fix the generation length to 512 and use a single temperature as reported in Appendix A.4.1, then vary only the beam size K to study how performance scales with K. We have made this distinction explicit in the revised manuscript.
> >
> > To address the concern that using an “optimal” temperature might favor OTS, we also perform a temperature-sensitivity study on GSM8K with generation_length = 128. We sweep T = {0.2, 0.5, 0.8, 1.0} and find that OTS achieves higher accuracy than low-confidence remasking at every temperature (see table below). This consistent ranking across all tested temperatures indicates that OTS’s gains are robust to the choice of decoding temperature.
> >
> > | Temperature | Low-confidence | OTS   |
> > |-------------|----------------|-------|
> > | T = 0.2     | 69.98          | 71.95 |
> > | T = 0.5     | 70.36          | 74.45 |
> > | T = 0.8     | 68.16          | 73.31 |
> > | T = 1.0     | 67.17          | 73.16 |
> >
> > > **W4: “The paper does not mention the extra inference overhead introduced by the joint search algorithm, which is important for evaluating practicality.”**
> >
> > We thank the reviewer for highlighting the importance of inference overhead in evaluating practicality. In Section 4.1, we briefly noted that, for each beam, “instead of searching at every denoising step—which would incur an O(K·|t|) overhead for |t| steps—we perform the search expansion only at the boundaries between contiguous blocks of tokens. This reduces the overhead to O(K·|b|), where |b| is the total number of blocks, making the search tractable.” We agree this deserves a more explicit treatment.
> >
> > Let S be the number of diffusion steps, L the generation length, K the beam size, and B the number of blocks. Vanilla DLM inference has time complexity O(S·L). A naive version of OTS that searches at every step would cost O(S·K²·L), because we maintain K beams and roll out K candidates per beam. Our actual OTS only searches at block boundaries, incurring B·K²·L extra forward passes, so the total NFE (number of forward evaluations) is approximately S·K·L (K independent denoising trajectories) plus B·K²·L (likelihood evaluations during search). In the main table setting where S = L/2 and B = L/32, this simplifies to NFE(OTS) = (L²·K)/2 + (K²·L²)/32. With our typical K ≈ 4, NFE(OTS) ≈ 2.5·L², which is comparable to the NFE of majority voting with 5 samples under the same diffusion schedule: NFE(MV, 5 samples) = S·5·L = 2.5·L². In other words, OTS offers structured joint search at roughly the same compute as a standard 5-sample majority-vote baseline.
> >
> > To further ground this in practical terms, we measured wall-clock time on the Countdown dataset, averaging over all problems and comparing low-confidence remasking, majority voting (5 samples), and OTS with 4 beams:
> >
> > | Method / Generation length     | 64    | 128   | 256   | 512    |
> > |---------------------|-------|-------|-------|--------|
> > | Low-confidence remasking       | 1.55  | 3.19  | 6.60  | 14.52  |
> > | + Majority-voting (5 samples)  | 7.73  | 15.94 | 32.99 | 72.59  |
> > | Order-Token Search (4 beams)   | 3.52  | 7.46  | 16.64 | 40.41  |
> >
> > These measurements show that OTS is roughly 2–3× slower than a single low-confidence run, but about 2× faster than naively sampling 5 times for majority voting at the same generation length, while delivering substantially higher accuracy. We have added this complexity discussion and the wall-time comparison in the updated manuscript to make the practical overhead and trade-offs of OTS clear.

---

> > > ### Author Response · Authors · 2025-11-20
> > >
> > > > **Q1: “In Figure 1, the legend contains a typo: ‘within a forwar’”**
> > >
> > > We thank the reviewer for pointing out this detail. We have fixed it in the updated manuscript.
> > >
> > > > **Q2: “How is Order-Token Search integrated with full diffusion generation (rather than block diffusion)? Is it applied only to per-block, and does it scale with the full diffusion length?”**
> > >
> > > We thank the reviewer for raising this clarification. Order-Token Search (OTS) is defined at the level of the full diffusion trajectory, not as an algorithm that only operates on isolated blocks. As described in Section 4.1, we start from a fully masked sequence and run the standard MDM denoising schedule for all diffusion steps on each beam; OTS only intervenes at user-chosen intervals (s, t) where we branch and prune candidates, while beams always represent full-length sequences. To keep this search tractable, in our experiments we instantiate OTS with block diffusion, which partitions the L positions into B contiguous blocks and triggers search at block boundaries. This is purely an efficiency device: the likelihood in Eq. 2 is always computed from the model’s full-sequence prediction x_0, and the score accumulates over the entire generation path, not per-block in isolation. In principle, one can recover “full-length” OTS by choosing a single block that spans the entire sequence (i.e., block_size = generation_length) and setting the search interval to control how often candidate expansions are performed; in that configuration, OTS evaluates and prunes candidates based on their likelihood over the whole sequence, while scaling with sequence length and diffusion steps as detailed in our complexity analysis. We will clarify this flexibility of OTS with respect to block size and search interval in the revised version.

---

> > > > ### Comment · Reviewer_34SD · 2025-11-25
> > > >
> > > > Thank you for your detailed response. I updated my score.

---

> > > > > ### Author Response · Authors · 2025-11-26
> > > > >
> > > > > Thank you for taking the time to re-evaluate our work and update your score. We really appreciate your careful feedback and the chance to clarify our contribution.

---

### Author Response · Authors · 2025-11-20

> **Expanded baselines and additional backbone model**


In response to requests for more baseline and backbone comparisons, we have substantially expanded our decoding baselines and backbones; please see the updated Table 1 in the revised manuscript, as well as the sampled result reported in the two tables below.


On the baseline side, we now include four stronger decoding strategies—**low-confidence + majority-voting**, **AR**, **AR + majority-voting**, and **AR + beam-search**. These additions primarily address Reviewer TD2K’s requests for stronger confidence-style and autoregressive baselines under comparable compute and Reviewer 34SD’s request for a beam-search counterpart to the AR baseline.


On the backbone side, following Reviewer RQJW’s concern about relying on a single model, we now evaluate OTS on both LLaDA-instruct and the RL-tuned LLaDA-1.5. Together with added baselines, the result addresses Reviewer HT8y’s concern that the extent of the experiments seemed lacking to prove significance.


As shown in the “All” (avg of 4 benchmarks), “MATH500-Avg”, and “Countdown-Avg” columns of the two tables below, OTS remains the best method on both backbones: for LLaDA, OTS achieves **35.2** in the All column, with **32.8** on MATH500-Avg and **28.4** on Countdown-Avg, outperforming the strongest non-OTS baselines (e.g., All = 33.3 with AR + beam-search). For LLaDA-1.5, OTS reaches **36.7** in the All column, with **33.8** on MATH500-Avg and **28.0** on Countdown-Avg, again strictly better than the strongest non-OTS baselines (All = 35.0, MATH500-Avg = 31.3, Countdown-Avg = 22.5 with Low-conf + MV). Thus, even with these stronger, compute-matched baselines and an additional backbone, OTS is still the overall top-performing decoding strategy.


**Table R1 (LLaDA backbone)**


| Method             | All |  |      |      |      |   MATH500   | |      |      |      |   Countdown   |
|--------------------|-----|---------|------|------|------|------|-----------|------|------|------|------|
|                    |     | L=64    | L=128 | L=256 | L=512 | Avg | L=64      | L=128 | L=256 | L=512 | Avg |
| Low-confidence     | 31.1| 21.2    | 26.0  | 32.4  | 36.2  | 29.0| 25.8      | 20.7  | 19.5  | 16.0  | 20.5 |
| Low-conf + MV      | 32.5| 20.2    | 27.4  | 35.0  | 36.2  | 29.7| 22.7      | 23.8  | 18.4  | 18.0  | 20.7 |
| Random + MV        | 28.8| 17.2    | 26.2  | 31.8  | 31.8  | 26.8| 6.3       | 15.2  | 14.1  | 15.2  | 12.7 |
| Order-Token Search | **35.2** | 22.4 | 30.4  | 36.0  | 42.4  | **32.8** | 27.7 | 34.4  | 26.2  | 25.4  | **28.4** |
| AR                 | 28.8| 18.8    | 23.4  | 27.4  | 34.4  | 26.0| 10.6      | 12.9  | 13.3  | 14.1  | 12.7 |
| AR + MV            | 31.0| 17.4    | 23.0  | 32.2  | 39.9  | 28.1| 10.2      | 13.3  | 11.3  | 13.7  | 12.1 |
| AR + beam-search   | 33.3| 22.2    | 26.6  | 35.4  | 39.8  | 31.0| 18.4      | 23.1  | 21.5  | 21.9  | 21.2 |




**Table R2 (LLaDA-1.5 backbone)**


| Method             | All     | |        |        |        |   MATH500   | |        |        |        |  Countdown    |
|--------------------|---------|---------|--------|--------|--------|------|-----------|--------|--------|--------|------|
|                    |         | L=64    | L=128  | L=256  | L=512  | Avg  | L=64      | L=128  | L=256  | L=512  | Avg  |
| Low-confidence     | 32.3    | 20.2    | 26.4   | 32.5   | 36.2   | 28.8 | 19.8      | 19.7   | 17.9   | 21.8   | 19.8 |
| Low-conf + MV      | 35.0    | 21.2    | 30.0   | 34.8   | 39.3   | 31.3 | 20.7      | 23.8   | 20.3   | 25.4   | 22.5 |
| Random + MV        | 30.2    | 22.4    | 26.2   | 31.0   | 30.4   | 27.5 | 5.5       | 16.4   | 9.0    | 14.5   | 11.4 |
| Order-Token Search | **36.7**| 24.4    | 30.8   | 37.4   | 42.4   | **33.8** | 27.7  | 31.3   | 23.8   | 29.3   | **28.0** |
| AR                 | 30.3    | 17.2    | 23.5   | 31.2   | 35.0   | 26.7 | 12.3      | 15.2   | 14.5   | 15.7   | 14.4 |
| AR + MV            | 32.2    | 18.4    | 26.4   | 33.6   | 37.9   | 29.1 | 12.5      | 17.2   | 14.8   | 16.0   | 15.1 |
| AR + beam-search   | 33.5    | 19.0    | 26.4   | 35.2   | 38.8   | 29.9 | 14.8      | 21.1   | 16.0   | 20.3   | 18.1 |

---

### Author Response · Authors · 2025-11-20

> **Core contribution: structured search over orders and tokens in DLMs**


To help reviewers better assess the novelty and significance of our work, we summarize here what is fundamentally new and why it matters for diffusion language models. At a high level, our central contribution is to treat DLM decoding not just as token prediction, but as *trajectory selection* through a space of reasoning orders. We formalize each problem as inducing a latent graph of logical dependencies, and view a DLM decoding trace as one concrete traversal of this graph: at every denoising step the model decides both **which position to update** and **which token to place there**. Standard MDM training only supervises token predictions under random remasking, so the distribution over trajectories is determined entirely by the inference-time remasking rule. Our analysis shows that low-confidence remasking collapses this trajectory space to a single greedy path that boosts pass@1 but hurts pass@k by restricting exploration, while random remasking explores many more orders and improves pass@k at the cost of weaker single-sample accuracy.


Building on this perspective, our algorithmic contribution is **Order–Token Search (OTS)**, a decoding procedure that performs *structured search* over both generation orders and token choices rather than committing to a single heuristic traversal. OTS uses the parallel denoising capability of MDMs to expand multiple candidate sequences in each search interval, scores them with a stable sequence-level likelihood estimator, and prunes to the most promising trajectories. Empirically, this joint order–token search reconciles the pass@1 vs. pass@k trade-off: across GSM8K, MATH500, and Countdown, OTS consistently improves pass@1 over low-confidence remasking while matching or surpassing the gains of fully post-trained d1-LLaDA with diffu-GRPO—using only test-time search instead of additional training. Together, the conceptual framing of reasoning as trajectory selection and the OTS decoding algorithm constitute the main contribution of our work, and we have incorporated this perspective explicitly into the Introduction of the revised manuscript.

---

### Author Response · Authors · 2025-12-03

We thank all reviewers for their valuable feedback and are encouraged that they found our motivation clear and the study of existing diffusion LM decoding methods intuitive and meaningful (34SD, RQJW, TD2K, HT8y). Reviewers highlighted the strong empirical gains across benchmarks (34SD, RQJW, TD2K).

Across the four reviews, the main concerns were about (i) fairness and strength of baselines, (ii) compute overhead and test-time scaling, (iii) robustness/interpretability of our scoring scheme, and (iv) whether our “reasoning” claims are supported beyond a single case study. During the rebuttal, we directly addressed reviewers’ questions on each of these points through clarifications and targeted additional analysis. As things stand, we have one positive reviewer who is explicitly leaning toward increasing their score, one initially more negative reviewer who has already updated their score upward, and two other reviewers who did not have the opportunity to update before the discussion period closed.

**Reviewer 34SD** focused on diversity controls, missing AR + beam-search baselines, the apparent discrepancy in Countdown numbers between an ablation table and the main results, and the lack of explicit overhead discussion. We clarified that all pass@k experiments already use a shared temperature across methods, added AR + beam search and other strong AR baselines under matched compute, explained the different experimental configurations behind the two Countdown numbers, and provided both NFE-based complexity analysis and wall-clock timing showing OTS uses similar compute to a 5-sample majority-vote baseline but achieves higher accuracy. After this, **34SD explicitly updated their score upward**.

**Reviewer RQJW** asked for FLOP-matched scaling curves, more than one backbone, and evidence that our block-level likelihood score is stable across block sizes. In response, we derived NFE expressions, added a test-time scaling plot (accuracy vs NFE) comparing OTS against majority voting, ran OTS on an additional RL-tuned backbone (LLaDA-1.5) with consistent gains, and presented a block-size sweep showing a broad plateau of strong performance, supporting our claim that block size is mainly an efficiency knob. In their follow-up, **RQJW stated they are leaning toward accepting the work and would like to update their rating**, but the discussion period closed shortly after, so the rating could not be formally adjusted.

**Reviewers TD2K and HT8y** raised related concerns about whether we truly go beyond “spending more compute,” whether additional baselines (AR, AR + MV, low-conf + MV with Gumbel) would close the gap, how our notion of “reasoning” is operationalized, and whether improvements are confined to Countdown. We responded by (i) emphasizing that OTS is designed to convert multi-sample potential into higher pass@1, (ii) adding AR, AR + MV, AR + beam search, and strengthened majority-voting baselines with shared temperature/Gumbel sampling under carefully matched NFE, where OTS still dominates, (iii) explicitly framing reasoning as trajectory selection over latent dependency graphs and OTS as structured joint search over orders and tokens, and (iv) highlighting that OTS matches or surpasses diffu-GRPO-level gains on three of four benchmarks using inference only. Due to the early close of the discussion period, **TD2K and HT8y did not yet have the chance to re-engage with these clarifications or adjust their scores**, but we hope our expanded results and explanations will be helpful in your overall assessment.

---

### Meta-Review · Area_Chair_HpmD · 2026-01-08

**Summary:**

This submission proposes a new decoding algorithm for diffusion language models that jointly searches over token orderings and token selections to improve reasoning performance. All reviewers raised two main concerns:
* inadequate experiments to demonstrate the significance of the performance improvement. In the rebuttal, the authors included experiments with more benchmarks and backbone models.
* the computational cost of the proposed decoding algorithm is not thoroughly discussed and compared against alternatives. In the revised manuscript, the authors added such discussion in Appendix A, with some preliminary wall-clock time comparison.

In addition, reviewers TD2K and 34SD raised questions regarding the interpretation of Figure 2, and reviewer HT8y mentioned the limitation of the case study.

Overall, the area chair believes that the authors' rebuttal partially addressed the major concerns, hence placing the submission on the borderline. However, the additional experimental results have not been thoroughly checked by peer reviews.

**Reviewer Concerns:**

See above.

**Reviewer Scores:**

Reviewer 34SD updated the rating towards acceptance after the discussion. Reviewer RQJW confirmed the positive evaluation.
Reviewers TD2K and HT8y did not update their reviews in the discussion period; the authors' response partially addressed their concerns on the lack of experimental comparison and discussion on computational overhead, and hence the they might increase their ratings to borderline.

---

### Decision · Program_Chairs · 2026-01-26

Reject